# A Local Polyak-Łojasiewicz and Descent Lemma of Gradient Descent for Overparameterized Linear Models

## Abstract

Most prior work on the convergence of gradient descent (GD) for overparameterized neural networks relies on strong assumptions on the step size (infinitesimal), the hidden-layer width (infinite), or the initialization (large, spectral, balanced). Recent work relaxes these assumptions for two-layer linear networks trained with the squared loss. In this work, we derive a linear convergence rate for training two-layer linear neural networks with GD for general losses and under relaxed assumptions on the step size, width, and initialization. A key challenge in deriving this result is that classical ingredients for deriving convergence rates for nonconvex problems, such as the Polyak-Łojasiewicz (PL) condition and Descent Lemma, do not hold globally for overparameterized neural networks. Here, we prove that these two conditions hold locally with constants that depend on the weights. Then, we provide bounds on these local constants, which depend on the initialization of the weights, the current loss, and the PL and smoothness constants of the non-overparameterized model. Based on these bounds, we derive a linear convergence rate for GD that can be shown to be asymptotically almost optimal w.r.t. the non-overparametrized GD. Our convergence analysis not only improves upon prior results but also suggests a better choice for the step size, as verified through our numerical experiments.

## 1 Introduction

Neural networks have shown great empirical success in many real-world applications, such as computer vision (He et al., 2016) and natural language processing (Vaswani et al., 2017). However, our theoretical understanding of why neural networks work so well is still scarce. One unsolved question is why neural networks trained via vanilla gradient descent (GD) enjoy fast convergence despite that their loss landscape is non-convex. Recent work has focused on deriving convergence rates for overparameterized neural networks. However, most prior work on the linear convergence of GD for overparametrized neural networks requires strong assumptions on the step size (infinitesimal), width (infinitely large), initialization (large, spectral), or restrictive choices of the loss function (squared loss) (See Table 1 for details).

Recently, many works have pointed out that the strong assumptions listed above are unrealistic, and neural networks that satisfy these assumptions perform poorly in practice. For example, one line of work (Du et al., 2018b; Lee et al., 2019; Liu et al., 2022) studies the convergence of GD in the neural tangent kernel (NTK) regime which requires the networks to have large or infinite widths and large initialization. However, Chizat et al. (2019); Chen et al. (2022) show that the NTK regime prohibits feature learning, and the generalization performance of neural networks in this regime degrades substantially. To relax the assumptions on the width, some recent works have focused on linear networks trained via gradient flow (GF) where GF can be viewed as GD with infinitesimal step size. Despite the fact that the dynamics of GF are generally easy to analyze, neural networks are never trained with infinitesimal step size in practice, and the corresponding analysis on the convergence rate of GF rarely provides meaningful information about the discrete counterpart (GD). Moreover, Barrett & Dherin (2020); Smith et al. (2021) show the step sizes of GD have implicit regularization, and the effect of regularization vanishes as the step sizes decrease to zero, which suggests the assumption of infinitesimal step size is inconsistent with the practical setting of GD. Hence, there is a need for an

Table 1: Comparison with prior work

| | Work | Loss | Step Size | Width | Initialization |
|---|---|---|---|---|---|
| Nonlinear networks | (Du et al., 2018b; Lee et al., 2019; Jacot et al., 2018; Liu et al., 2022; Nguyen & Mondelli, 2020) | Squared loss | **Finite** | Large | Large |
| | (Mei et al., 2018; Chizat & Bach, 2018; Sirignano & Spiliopoulos, 2020; Ding et al., 2022) | Squared loss | Infinitesimal | Large | **General** |
| Linear networks | (Saxe et al., 2013; Gidel et al., 2019; Tarmoun et al., 2021) | Squared loss | Infinitesimal | **Finite** | Spectral |
| | (Arora et al., 2018; Du et al., 2018a) | Squared loss | **Finite** | **Finite** | Large margin and small imbalance |
| | (Xu et al., 2023) | Squared loss | **Finite** | **Finite** | **General** |
| | This work | **General** | **Finite** | **Finite** | **General** |

analysis that establishes the convergence of neural networks trained using GD under more relaxed assumptions and with more accurate predictions of the actual rates of convergence.

## 1.1 MAIN CONTRIBUTION

In this work, we derive (asymptotically) tight linear convergence rates for GD on overparameterized two-layer linear networks with a general loss, finite width, finite step size, and general initialization. The main contributions of the paper are:

- We analyze the Hessian of two-layer linear networks and show that the optimization problem satisfies a local PL condition and local Descent Lemma, where we characterize the local PL constant and local smoothness constant along the descent direction around GD iterates[1] by their corresponding loss values and the singular values of the weight matrices.

- We show that when the step size satisfies certain constraints (not infinitesimal), the *imbalance*[2] of the network weights remains close to its initial value. Based on this property, we show that the local PL and smoothness constants can be bounded along the trajectory of GD, which leads to a linear convergence rate for GD. Moreover, our results cover GD with decreasing, constant, and increasing step sizes while prior work (Du et al., 2018a; Arora et al., 2018; Xu et al., 2023) only covers GD with decreasing and constant step sizes.

- We show that the local smoothness constant decreases along the GD trajectory under certain constraints on the step size, which suggests the optimization landscape gets more benign as the training proceeds. Based on this observation, we design an adaptive step size scheduler that accelerates the convergence (See Appendix G).

- Our analysis allows us to show that when GD iterates are around a global minimum, the difference between the local rate of convergence of the overparametrized model and the rate of the non-overparametrized model is up to one condition number of an operator (see §2.2 for definitions) which can be made arbitrarily close to one by proper initialization. Thus showing that the non-convexity induced by the overparametrization mildly affects convergence.

## 1.2 RELATED WORK

We now provide a detailed description of prior work in addition to the discussion above.

One line of work (Du et al., 2018b; Lee et al., 2019; Liu et al., 2022; Nguyen & Mondelli, 2020) studies the convergence of GD under the assumption that the width and initialization of neural networks are sufficiently large, which is also known as the neural tangent kernel (NTK) regime. Under these assumptions, the training trajectories of a neural network are governed by a kernel determined at initialization and the network weights stay close to their initial values. Such properties

---

[1]In the paper, we adopt the term *local smoothness constant* as a convenient shorthand to refer to the smoothness constant along the descent direction around GD.

[2]The *imbalance* is a quantity that measures the difference between the weights of two adjacent layers.

help them derive a linear convergence rate of GD. However, Chizat et al. (2019); Chen et al. (2022) show that the NTK regime prohibits feature learning, and the performance of neural networks in this regime degrades substantially. Another line of work (Mei et al., 2018; Rotskoff & Vanden-Eijnden, 2018a;b; Sirignano & Spiliopoulos, 2020) studies the convergence of GD in the (mean-field) limit of infinitely wide neural networks with the infinitesimal step size assumption, where the dynamics of network weights follow a partial differential equation. However, such analysis imposes strong assumptions on the width (infinite) and step size (infinitesimal), and the convergence is shown without an explicit bound on the rate.

To relax the assumptions on the width, step size, or initialization, many work focus on deriving convergence rates of gradient-based algorithms for neural networks with linear activation functions, based on the observation that linear networks exhibit similar nonlinear learning phenomena to those seen in simulations of nonlinear networks (Saxe et al., 2013). In the finite-width setting, most existing results consider linear networks trained using GF. GF can be seen as GD with infinitesimal step size, but its dynamics in this setting are generally easier to analyze. For example, Saxe et al. (2013); Tarmoun et al. (2021); Min et al. (2022) show that linear networks enjoy linear convergence under different assumptions on the initialization. However, all results require infinitesimal step size. In the finite step size regime, Arora et al. (2018); Du et al. (2018a); Xu et al. (2023) show the linear convergence of GD. Specifically, Arora et al. (2018); Du et al. (2018a) derive the linear convergence of GD when there is sufficient *margin* and small *imbalance* at initialization where the *margin* measures how close the initialization is to the global minimum. However, such initialization is impractical since commonly used random initialization schemes, such as Xavier initialization (Glorot & Bengio, 2010) and He initialization (He et al., 2015), both lead to a large *imbalance*. Recently, Xu et al. (2023) derive a convergence rate for GD under general initialization where there is either sufficient *imbalance* or sufficient *margin*. Moreover, they design an adaptive step size scheme that accelerates the convergence. However, the convergence rate of the adaptive step size in (Xu et al., 2023) only holds under stringent assumptions on some auxiliary constants which leads to a slower convergence rate. Despite the relaxed assumptions on the width, step size, or initialization in (Arora et al., 2018; Du et al., 2018a; Xu et al., 2023), all these works considers squared loss, and the analyses are based on the PL condition and Descent lemma of the non-overparameterized models and do not fully capture the optimization properties of linear networks.

A classical approach to deriving the convergence rate of GD for non-convex optimization problems is based on Descent lemma and PL condition Karimi et al. (2020), both of which are closely related to the Hessian. However, most results for neural networks focus on characterizing the structure of the Hessian and do not connect to the convergence rate of GD. For example, Sagun et al. (2017); Wei & Schwab (2019); Alain et al. (2019); Ghorbani et al. (2019); Sun et al. (2020) empirically study the evolution of eigenvalues of the Hessian during training. Some theoretical work mainly focuses on finding certain structures of the Hessian, such as low-rank (Singh et al., 2021; Wu et al., 2022) or characterization of the top eigenvalue under a constrained setting (Zhou & Liang, 2017). The only work that we are aware of that uses the properties of Hessian to derive the convergence rate of GD for overparameterized neural networks is (Liu et al., 2022), which shows that when the width is very large, the Hessian is almost constant during training. Under the assumption that the optimization problem satisfies the PL condition at initialization and the smoothness condition everywhere, they prove the linear convergence of GD. However, neural networks are not globally Lipschitz-smooth, and the statement that Hessian is almost constant is inconsistent with practical observations (Sagun et al., 2017; Ghorbani et al., 2019). Therefore, an analysis that fully exploits information of the neural network's Hessian in the derivation of the convergence rate is still missing.

**Notation.** We use lower case letters $a$ to denote a scalar, and capital letters $A$ and $A^\top$ to denote a matrix and its transpose. We use $\sigma_{\max}(A)$ and $\sigma_{\min}(A)$ to denote the largest and smallest singular values of $A$, $\|A\|_F$ and $\|A\|_2$ to denote its Frobenius and spectral norms, and $A[i,j]$ to denote its $(i,j)$-th element. For a function $f(Z)$, we use $\nabla f(Z) := \frac{\partial}{\partial Z} f(Z)$ to denote its gradient.

## 2 PRELIMINARIES

In this paper, we consider using the GD algorithm to solve the following optimization problem

$$\min_{W \in \mathbb{R}^{n \times m}} \ell(W), \tag{1}$$

and its overparametrized version

$$\min_{W_1 \in \mathbb{R}^{n \times h}, W_2 \in \mathbb{R}^{m \times h}} L(W_1, W_2) = \ell(W_1 W_2^\top). \tag{2}$$

We are mostly interested in solving Problem 2, which covers many problems, such as matrix factorization (Koren et al., 2009), matrix sensing (Chen & Chi, 2013), training linear neural networks (Arora et al., 2018; Du et al., 2018a; Xu et al., 2023). In particular, when $\ell(W) = \frac{1}{2}\|Y - XW\|_F^2$, where $X, Y$ are data matrices, Problem 2 corresponds to training a two-layer linear neural network with $n$ inputs, $h$ hidden neurons, $m$ outputs, and weight matrices $W_1$ and $W_2$ using the squared loss.

## 2.1 Convergence rate of GD for Problem 1

In this section, we review the analysis for deriving the convergence rate of GD for Problem 1.

We seek to derive the convergence rate of GD for Problem 1 with the following iterations,

$$W(t+1) = W(t) - \eta_t \nabla\ell(W(t)), \tag{3}$$

where we will use $\ell(t), \nabla\ell(t)$ as a shorthand for $\ell(W(t)), \nabla\ell(W(t))$ respectively.

Throughout the paper, we make the following assumptions.

**Assumption 2.1.** *The loss $\ell(W)$ is twice differentiable, $K$-smooth, and $\mu$-strongly convex .*

**Assumption 2.2.** $\min_W \ell(W) = 0$.

Assumption 2.1 ensures the solution to Problem 1 is unique. Assumption 2.2 is for the purpose of convenience and brevity of theorems in this work. This assumption can be relaxed (to have arbitrary $\ell^*$) without affecting the significance of our results. Moreover, one can have the following inequalities based on the above assumptions for arbitrary $W, V \in \mathbb{R}^{n \times m}$

$$\ell(V) \leq \ell(W) + \langle \nabla\ell(W), V - W \rangle + \frac{K}{2}\|V - W\|_F^2 \qquad \text{Smoothness inequality}, \tag{4}$$

$$\frac{1}{2}\|\nabla\ell(W)\|_F^2 \geq \mu\ell(W) \qquad\qquad\qquad \text{PL inequality}. \tag{5}$$

Since strong convexity implies PL condition, equation 5 holds under Assumption 2.1. In §3, we derive the convergence rate of Problem 2 based on the argument of the local PL condition. To be consistent, we highlight the role of the PL condition here. Moreover, the analysis in §2.1 remains applicable when $\mu$-strong convexity is relaxed to $\mu$-PL condition.

In (Polyak, 1963; Boyd & Vandenberghe, 2004), it was shown that whenever $0 < \eta_t < \frac{2}{K}$, the GD iteration equation 3 achieves linear convergence. The derivation is based on two ingredients: Descent lemma and the PL inequality where Descent lemma is derived from the smoothness inequality.

**Descent lemma.** Starting from the smoothness inequality in equation 4, one can substitute $(V, W)$ with the GD iterates $(W(t+1), W(t))$ to derive Descent lemma, i.e.,

$$\ell(t+1) \leq \ell(t) + \langle \nabla\ell(t), W(t+1) - W(t) \rangle + \frac{K}{2}\|W(t+1) - W(t)\|_F^2 = \ell(t) - (\eta_t - \frac{K\eta_t^2}{2})\|\nabla\ell(t)\|^2.$$

Based on the PL inequality in equation 5 and Descent lemma above, one can see there is a strict decrease in the loss at each GD step

$$\ell(t+1) \leq \ell(t) - (\eta_t - \frac{K\eta_t^2}{2})\|\nabla\ell(t)\|^2 \leq (1 - 2\mu\eta_t + \mu K\eta_t^2)\ell(t), \tag{6}$$

where the fact that $0 < \eta_t < \frac{2}{K}$, implies $0 < 1 - 2\mu\eta_t + \mu K\eta_t^2 < 1$. Moreover, the minimum descent rate in equation 6 is achieved when $\eta_t = \frac{1}{K}$, leading to the following linear convergence rate:

$$\ell(t+1) \leq \left(1 - \frac{\mu}{K}\right)\ell(t) \leq \left(1 - \frac{\mu}{K}\right)^{t+1}\ell(0). \tag{7}$$

**Tightness of the analysis.** The previous analysis guarantees a linear convergence rate for any arbitrary non-convex function that is $K$-smooth and satisfies the $\mu$-PL condition. Moreover, one can show that the rate in equation 7 is optimal in the sense that there exists a function that is $K$-smooth and satisfies the $\mu$-PL condition for which the bound on equation 7 is met with equality. Therefore, one would naturally be tempted to apply such an analysis to Problem 2. We will next show that overparameterization introduces several challenges that prevent this analysis from being readily applied.

## 2.2 CHALLENGES IN THE ANALYSIS OF CONVERGENCE OF PROBLEM 2 OPTIMIZED VIA GD

In this section, we first introduce GD with adaptive step size to solve Problem 2. Then, we discuss the main challenges in deriving the convergence rate for Problem 2 based on the analysis in §2.1.

**Overparametrized GD.** We consider using GD with adaptive step size $\eta_t$ to seek optimal solutions of Problem 2.

$$\begin{bmatrix} W_1(t+1) \\ W_2(t+1) \end{bmatrix} = \begin{bmatrix} W_1(t) \\ W_2(t) \end{bmatrix} - \eta_t \nabla L\big(W_1(t), W_2(t)\big), \tag{8}$$

where $\nabla L(W_1, W_2)$ is computed via the chain rule:

$$\nabla L(W_1, W_2) = \mathcal{T}(\nabla \ell(W); W_1, W_2) := \begin{bmatrix} \nabla \ell(W) W_2 \\ \nabla \ell(W)^\top W_1 \end{bmatrix}. \tag{9}$$

Here $\mathcal{T} : \mathbb{R}^{n \times m} \mapsto \mathbb{R}^{(n+m) \times h}$ is a weight-dependent linear operator that acts on $\nabla \ell(W)$. Thus, the gradient of $L$ in equation 9 can be viewed as a "skewed/scaled gradient" of $\ell$ that depends on $W_1, W_2$. It is this dependence on the weights $W_1, W_2$ that makes it impossible to globally guarantee that equation 4 and equation 5 hold, as shown next.

**Proposition 2.1** (Non-existence of global PL constant and smoothness constant). *Under mild assumptions, the PL inequality and smoothness inequality can only hold globally with constants $\mu_{over} = 0$ and $K_{over} = \infty$ for $L(W_1, W_2)$.*

The proof of the above proposition can be found in Appendix B.

The non-existence of global PL and smoothness constants in the over-parametrized models prevents us from using the same proof technique in §2.1 to derive the linear convergence of GD. In §3, we show that although these constants do not exist globally, we can characterize them along iterates of GD. Moreover, under proper choices of the step size of GD, the PL and smoothness constants can be controlled for all iterates of GD. Thus, the linear convergence of GD can be derived.

## 3 CONVERGENCE OF GD FOR PROBLEM 2

To deal with the challenges presented in §2.2, in §3.1 we propose a novel PL inequality and Descent Lemma evaluated on the iterates of GD for Problem 2. Next, based on the results in §3.1, in §3.2 we derive a convergence rate for GD that depends on the weights at initialization, the step size, $K$, and $\mu$. Moreover, in §3.2 we propose an adaptive step size scheduler that dynamically optimizes the rate to accelerate convergence.

Throughout the paper, we assume that the width satisfies $h \geq \min\{n, m\}$. This assumption ensures $\ell^* = L^*$ where $L^* = \min_{W_1, W_2} L(W_1, W_2)$, and thus solving Problem 2 yields the solution to Problem 1. When $h < \min\{n, m\}$, Problem 2 enforces a rank constraint on the product. Thus, $\min_W \ell(W)$ may not be equal to $\min_{W_1, W_2} L(W_1, W_2)$. We are therefore interested in studying Problem 2 under the assumption $h \geq \min\{n, m\}$ which is the same setting in (Arora et al., 2018; Du et al., 2018a; Xu et al., 2023).

### 3.1 LOCAL PL INEQUALITY AND DESCENT LEMMA FOR OVER-PARAMETRIZED GD

In §2.2, we saw that there does not exist a global PL constant or a global smoothness constant for Problem 2. However, to prove that GD converges linearly to a global minimum of Problem 2, it is sufficient for Descent lemma and PL inequality to hold for iterates of GD. The following theorem formally characterizes the local PL inequality and Descent lemma for Problem 2.

**Theorem 3.1** (Local Descent Lemma and PL condition for GD iterates). *At the $t$-th iteration of GD applied to the Problem 2, the Descent lemma and PL inequality hold with local smoothness constant $K_t$ and PL constant $\mu_t$ , i.e.,*

$$L(t+1) \leq L(t) - \big(\eta_t - \frac{K_t \eta_t^2}{2}\big)\|\nabla L(t)\|_F^2, \quad \frac{1}{2}\|\nabla L(t)\|_F^2 \geq \mu_t L(t). \tag{10}$$

*Moreover, if the step size $\eta_t$ satisfies $\eta_t > 0$ and $\eta_t K_t < 2$, then the following inequality holds*

$$L(t+1) \leq L(t)(1 - 2\mu_t\eta_t + \mu_t K_t \eta_t^2) := L(t)\rho(\eta_t, t), \tag{11}$$

*where*

$$\mu_t = \mu\sigma_{\min}^2(\mathcal{T}_t), \tag{12}$$

$$K_t = K\sigma_{\max}^2(\mathcal{T}_t) + \sqrt{2KL(t)} + 6K^2\sigma_{\max}(W(t))L(t)\eta_t^2 + 3K\sigma_{\max}^2(\mathcal{T}_t)\sqrt{2KL(t)}\eta_t, \tag{13}$$

*and we use $L(t)$ and $\mathcal{T}_t$ as shorthands for $L(W_1(t), W_2(t))$ and $\mathcal{T}(\,\cdot\,; W_1(t), W_2(t))$, resp.*

The proof of the above theorem can be found in Appendix D. Notice that $\mu_t, K_t$ are not actually constants since they vary w.r.t. the iteration index $t$. In this work, we adopt the convention to call them local PL and smoothness constants to be consistent with the terminology in §2.

In §2.1, we showed that as long as one chooses $\eta_t = \bar{\eta}$, with $0 < \bar{\eta} < \frac{2}{K}$, GD in equation 3 for Problem 1 achieves linear convergence, with an optimal rate $(1 - \frac{\mu}{K})$ given when $\eta_t = \frac{1}{K}$. However, we argue Theorem 3.1 does not imply linear convergence of overparametrized GD even though there always exists sufficiently small $\eta_t > 0$ such that $\eta_t K_t < 2$. The difference is due to the fact that $\mu_t$ and $K_t$ are changing w.r.t. the iterations. Specifically, if $\lim\limits_{t\to\infty} \frac{\mu_t}{K_t} = 0$, one has $\lim\limits_{t\to\infty} \inf_{0 < \eta_t < \frac{2}{K_t}} \rho(\eta_t, t) = 1$. Thus, equation 11 does not necessarily imply that the product of the per-iterate descent $\Pi_{l=0}^t \rho(\eta_l, l)$ goes to zero.

**Towards linear convergence.** Nevertheless, if there exists $\eta_t > 0$ that can simultaneously satisfy the constraint $\eta_t K_t < 2$ and the uniformly bound $1 - 2\mu_t\eta_t + \mu_t K_t \eta_t^2 \leq \bar{\rho} < 1$, for all $t$, one can expect the linear convergence

$$L(t+1) \leq \rho(\eta_t, t)L(t) \leq \bar{\rho}L(t) \leq \bar{\rho}^{t+1}L(0). \tag{14}$$

Guaranteeing a uniform bound as in equation 14, requires one to keep track and control the evolution of $W(t), \mathcal{T}_t, \eta_t$ and $L(t)$. In the next section, we will address these issues. For the time being, we focus next on how the $\mu_t, K_t$ in Theorem 3.1 depend on the $\mu, K, L(t), \eta_t$ and the current weights.

**Characterization of $\mu_t, K_t$.** Theorem 3.1 shows how overparametrization affects the local PL constant and smoothness constant, i.e., $\mu_t, K_t$, via a time-varying linear operator $\mathcal{T}_t$. Specifically, the PL constant in equation 12 is the PL constant of $\ell(W)$, i.e., $\mu$, scaled by $\sigma_{\min}^2(\mathcal{T}_t)$. Moreover, the smoothness constant in equation 13 consists of two parts. The first one is $K\sigma_{\max}^2(\mathcal{T}_t)$, which represents the smoothness constant of $\ell(W)$, i.e., $K$, scaled by $\sigma_{\max}^2(\mathcal{T}_t)$. The rest of the terms decrease to zero as the loss $L(t)$ approaches zero.

In the next section, we will show that proper choice of initialization and step sizes $\eta_t$ does indeed lead to linear convergence of overparametrized GD.

## 3.2 Linear Convergence of Problem 2 with GD

In this section, we first state a theorem which shows that GD in equation 8 converges linearly to a global minimum of Problem 2 (See Theorem 3.2) under certain constraints on $\eta_t$ and the initialization. Then, based on the convergence rate in Theorem 3.2, we propose an adaptive step size scheduler that accelerates the convergence. Finally, we present a sketch of the proof of Theorem 3.2 to highlight the technical novelty and implications of the theorem. We refer the reader to Table 2 for the definition of various quantities appearing in this section.

We now present our main result on the linear convergence of GD for Problem 2.

**Theorem 3.2** (Linear convergence of GD for Problem 2). *Assume the GD algorithm in equation 8 is initialized such that $\alpha_1 > 0$. Then there exists $\eta_{\max} > 0$ such that for all $\eta_0, \eta_t$ that satisfies $0 < \eta_0 < \eta_{\max}$ and*

$$\eta_0 \leq \eta_t \leq \min\left((1 + \eta_0^2)^{\frac{t}{2}}\eta_0, \frac{1}{K_t}\right), \tag{15}$$

*one can derive the following bound for each iteration*

$$L(t+1) \leq L(t)\bar{\rho}(\eta_t, t) \leq L(t)\bar{\rho}(\eta_0, 0). \tag{16}$$

Table 2: Notation

| Symbol | Definition | Symbol | Definition |
|--------|-----------|--------|-----------|
| $D(t)$ | $W_1^\top(t)W_1(t) - W_2^\top(t)W_2(t)$ | $\underline{\Delta}$ | $\max(\lambda_n(D(0)),0) + \max(\lambda_m(-D(0)),0)$ |
| $\lambda_-$ | $\max(\lambda_{\max}(-D(0)),0)$ | $\alpha_1$ | $\dfrac{-\Delta_+ - \Delta_- + \sqrt{(\Delta_+ + \underline{\Delta})^2 + 4\beta_1^2} + \sqrt{(\Delta_- + \underline{\Delta})^2 + 4\beta_1^2}}{2}$ |
| $\lambda_+$ | $\max(\lambda_{\max}(D(0)),0)$ | $\alpha_2$ | $\dfrac{\lambda_+ + \sqrt{\lambda_+^2 + 4\beta_2^2}}{2} + \dfrac{\lambda_- + \sqrt{\lambda_-^2 + 4\beta_2^2}}{2}$ |
| $\Delta_+$ | $\lambda_+ - \max(\lambda_n(D(0)),0)$ | $\beta_1$ | $\max\!\big(0, \sigma_{\min}(W^*) - \sqrt{\tfrac{K}{\mu}}\|W(0) - W^*\|_F\big)$ |
| $\Delta_-$ | $\lambda_- - \max(\lambda_m(-D(0)),0)$ | $\beta_2$ | $\sigma_{\max}(W^*) + \sqrt{\tfrac{2}{\mu}L(0)}$ |

*Moreover, based on equation 16, GD algorithm in equation 8 converges linearly*

$$L(t+1) \le L(0)\bar{\rho}(\eta_0,0)^{t+1} \,, \tag{17}$$

*where*

$$\bar{\rho}(\eta_t,t) = 1 - 2\bar{\mu}\eta_t + \bar{\mu}\bar{K}_t\eta_t^2 \,, \quad \bar{\mu} = \mu\big[\alpha_1 + 2\alpha_2\big(1 - \exp(\sqrt{\eta_0})\big)\big]\,, \quad \Delta = (1+\eta_0^2)\bar{\rho}(\eta_0,0)\,,$$

$$\bar{K}_t = \sqrt{2KL(0)\bar{\rho}(\eta_0,0)^t} + 6K^2\beta_2 L(0)\eta_0^2\Delta^t + K\exp(\sqrt{\eta_0})\alpha_2\big[1 + 3\sqrt{2KL(0)\Delta^t}\eta_0\big]\,.$$

The proof of the above theorem is presented in Appendix E. The above theorem states GD enjoys linear convergence for Problem 2 under the assumptions that $\alpha_1 > 0$ and certain constraints on $\eta_t$. We make the following remarks:

**Conditions on the initialization for linear convergence.** From Theorem 3.2, we see that if the initialization $\{W_1(0), W_2(0)\}$ satisfies $\alpha_1 > 0$, then GD converges linearly with an appropriate choice of the step size. The constraints on $\eta_0$ ensure that $\bar{\mu} > 0$. Thus, when $0 < \eta_0 < \frac{2}{\bar{K}_0}$, one has $0 < \bar{\rho}(\eta_0,0) < 1$. The assumptions on $\alpha_1$ has been studied in Min et al. (2022) where the authors show that $\alpha_1 > 0$ when there is either 1) sufficient imbalance $\underline{\Delta} > 0$ or 2) sufficient margin $\beta_1 > 0$, where $\underline{\Delta}, \beta_1$ is defined in Table 2. In Appendix F, we present two conditions that ensure $\alpha_1 > 0$. Please see Appendix F for a detailed proof and discussions.

**Evolution of smoothness constant.** One unique feature in our Theorem is the time-varying upper bound $\bar{K}_t$ on the local smoothness constant $K_t$ along GD iterates. The constraints on $\eta_0$ ensures that $0 < \bar{\rho}(\eta_0,0), \Delta < 1$. Thus, $\bar{K}_t$ monotonically decrease to $K\exp(\sqrt{\eta_0})\alpha_2$ w.r.t. $t$. The fact that $\bar{K}_t$ is monotonically decreasing w.r.t. $t$ suggests that the local optimization landscape gets more benign as the training proceeds. Thus in order to achieve a fast rate of convergence, there is a need for a time-varying choice of step size that adapts to the changes in the local smoothness constant $K_t$ (because theoretically, the optimal choice is $\eta_t = \frac{1}{K_t}$, based on equation 11). In Zhang et al. (2020), the authors show empirically that the global smoothness condition does not hold in deep neural networks, and keeping track of the smoothness constant is important to understand the acceleration of optimization methods, such as gradient clipping. The characterization of the smoothness constant, i.e. $K_t, \bar{K}_t$, for two-layer linear networks can be the first step to help us understand the acceleration of optimization algorithms for deep neural networks.

**Requirement on the step size.** We have mentioned in the previous remark that a time-varying step size could be beneficial for convergence. However, prior analyses (Arora et al., 2018; Du et al., 2018a; Xu et al., 2023) are all restricted to a constant or decaying step size. The main reason is that one requires a uniform spectral bound on $\mathcal{T}_t$ and $W(t)$ throughout the entire GD trajectory to establish linear convergence and such a uniform bound has only been shown under a constant or decaying step size. In our analysis, we show a similar spectral bound can be obtained even with a growing step size (See Lemma 3.1 in Section 3.3), as long as $\eta_t \le (1+\eta_0^2)^{\frac{t}{2}}\eta_0$, but not too much $\eta_t \le \frac{1}{\bar{K}_t}$ (ensures a sufficient decrease in the loss at every iteration). The first bound diverges to infinity exponentially fast, and the second bound has a growing lower bound $\frac{1}{\bar{K}_t}$ which monotonically increases to $\frac{1}{K\exp(\sqrt{\eta_0})\alpha_2}$. Thus, initially, the step size is restricted to $[\eta_0, (1+\eta_0^2)^{\frac{t}{2}}\eta_0]$ thus not much larger than the initial step $\eta_0$. As the training goes on, the binding constraint becomes $\eta_t \le \frac{1}{\bar{K}_t}$, suggesting that GD can take a step that achieves the theoretically largest descent in the loss.

**Local rate of convergence .** In Lemma 3.2, we show $L(t+1) \le L(t)\bar{\rho}(\eta_t,t)$. When $t$ is sufficiently large, or equivalently around any global minimum of Problem 2, the $\bar{\rho}(\eta_t,t)$ takes a simple form, and

the optimal rate of convergence is achieved as (via a proper choice of $\eta_t$)

$$\min_{\eta_t} \bar{\rho}(\eta_t, t) = \min_{\eta_t} 1 - 2\bar{\mu}\eta_t + \bar{\mu}K \exp(\sqrt{\eta_0})\alpha_2\eta_t^2 = 1 - \frac{\mu}{K} \cdot \frac{\alpha_1 + 2\alpha_2\left(1 - \exp(\sqrt{\eta_0})\right)}{\exp(\sqrt{\eta_0})\alpha_2}.$$

Notice the optimal local rate of convergence can be arbitrarily close to $1 - \frac{\mu}{K} \cdot \frac{\alpha_1}{\alpha_2}$ as $\eta_0$ decreases.

Table 3: Comparison of convergence rates between prior work and our work.

| | loss | step size | initialization | local rate of convergence |
|---|---|---|---|---|
| Arora et al. (2018) | squared loss | constant | $D(0) \approx 0, \beta_1 > 0$ | $1 - \Omega(\frac{\mu\alpha_1^2}{K\alpha_2^2})$ |
| Du et al. (2018a) | squared loss | decreasing | $D(0) \approx 0, \beta_1 > 0$ | no explicit rate |
| Xu et al. (2023) | squared loss | constant | $\alpha_1 > 0$ | $1 - \Omega(\frac{\mu\alpha_1^2}{K\alpha_2^2})$ |
| our work | general | adaptive | $\alpha_1 > 0$ | $1 - \Omega(\frac{\mu\alpha_1}{K\alpha_2})$ |

**Detailed Comparison with SOTA.** We compare our results with prior works studying the same problem Arora et al. (2018); Du et al. (2018a); Xu et al. (2023) (See Table 3). Moreover, we present a detailed discussion on the difference of proof techniques used in this work and prior work, and how it leads to different convergence rates. Please see Appendix C for details.

**Choices of the step size.** Recall that for non-overparamterized GD, we have $\ell(t+1) \leq (1 - 2\mu\eta_t + \mu K\eta_t^2)\ell(t)$, there exists an optimal choice of $\eta_t^* = \frac{1}{K}$ that minimize the theoretical upper bound on $\ell(t+1)$. In Theorem 3.1 and Theorem 3.2, we show $L(t+1) \leq h(\eta_t, t)L(t)$ under certain conditions on $\eta_t$ where $h(\eta_t, t) \in \{\rho(\eta_t, t), \bar{\rho}(\eta_t, t)\}$. It is natural to use a similar approach to select step size at each iteration. To achieve the optimal step size, it suffices to minimize the upper bound on $L(t+1)$ to achieve the most decrease at each iteration. The difference is that we have a time-varying upper bound on $L(t+1)$ thus the minimizer $\eta_t^*$ depends on time, and our choice of $\eta_t^*$ must respect our constraint on step size in equation 15. This leads to the following choice for $\eta_t^*$

$$\eta_t^* = \underset{\eta_t \leq \min\{(1+\eta_0^2)^{t/2}\eta_0, \frac{1}{K_t}\}}{\arg\min} h(\eta_t, t). \tag{18}$$

Since $\rho(\eta_t, t), \bar{\rho}(\eta_t, t)$ are quadratic in terms of $\eta_t$, so $\eta_t^*$ takes the following closed-form solutions depending on which upper bound to use:

$$\eta_t^* = \begin{cases} \min\left((1 + \eta_0^2)^{t/2}\eta_0, \frac{1}{K_t}\right) & \text{if } h(\eta_t) = \rho(\eta_t, t), \\ \min\left((1 + \eta_0^2)^{t/2}\eta_0, \frac{1}{K_t}\right) & \text{if } h(\eta_t) = \bar{\rho}(\eta_t, t). \end{cases} \tag{19}$$

The above choices of the adaptive step sizes satisfy the constraints in Theorem 3.2, so they both guarantee linear convergence for over-parametrized GD. Moreover, such choices of $\eta_t$ give us the following theoretical bound on $L(t+1)$, i.e.,

$$L(t+1) \leq L(0) \prod_{k=1}^{t} h(\eta_k^*, k). \tag{20}$$

In Appendix G, we provide numerical verification of the close alignment between the theoretical bounds stated above and the actual convergence rate. We also observe an accelerated convergence when employing the step sizes specified in equation 18 compared with the one proposed in Xu et al. (2023) and Backtracking line search. We refer the readers to Appendix G for simulation results.

### 3.3 PROOF SKETCH OF THEOREM 3.2

In this section, we provide a proof sketch of Theorem 3.2 that highlights the technical novelty and implications of our results.

To show there exists a $0 < \bar{\rho} < 1$ such that $\rho(\eta_t, t) \leq \bar{\rho}$ holds for all $t$, we use the following two-step approach in a similar way as it was done in (Xu et al., 2023).

**Step one: uniform spectral bounds for $\mathcal{T}_t, W_t$.** First, we show when $\eta_t$ is controlled, one has the following uniform spectral bounds on $\mathcal{T}_t$ and $W(t)$.

**Lemma 3.1** (Uniform spectral bounds on $\mathcal{T}_t, W(t)$.)**.** *Under the same assumption and constraints in Theorem 3.2, one has the following uniform spectral bounds on $\mathcal{T}_t, W(t)$*

$$\alpha_1 + 2\alpha_2\big(1 - \exp(\sqrt{\eta_0})\big) \leq \sigma_{\min}^2(\mathcal{T}_t) \leq \sigma_{\max}^2(\mathcal{T}_t) \leq \alpha_2 \exp(\sqrt{\eta_0})\,, \tag{21}$$

$$\beta_1 \leq \sigma_{\min}(W(t)) \leq \sigma_{\max}(W(t)) \leq \beta_2\,. \tag{22}$$

Similar results have been derived in (Xu et al., 2023) where the authors show uniform spectral bounds of $\mathcal{T}_t, W_t$ for constant step size GD. Our proof strategy is similar to (Xu et al., 2023) which relies on the fact that when the GD enjoys linear convergence, the change of imbalance during the training is small. For constant step size GD, we can characterize the change using the step size $\eta$ and the convergence rate $\bar{\rho}$, i.e., $\|D(t) - D(0)\|_F \leq \mathcal{O}(\frac{\eta^2}{1-\bar{\rho}})$. In this work, we discover when we allow step size to grow but not too fast, i.e., $\eta_t \leq (1 + \eta_0^2)^{\frac{t}{2}}\eta_0$, we can still control the change of imbalance, i.e., $\|D(t) - D(0)\|_F \leq \mathcal{O}(\frac{\eta_0^2}{1-\Delta})$. This observation helps us derive the uniform spectral bounds for $\mathcal{T}_t, W(t)$ while allowing the step size to grow.

**Step two.** Second, we employ an induction-based argument to prove that based on Lemma 3.1, one can show $\mu \geq \bar{\mu}, K_t \leq \bar{K}_t$ and $L(t)$ converges linearly with the rate $\bar{\rho}(\eta_0, 0)$.

**Lemma 3.2** (Induction step to show $\mu_t, K_t$ is bounded and $L(t)$ converges linearly.)**.** *Under the same assumption and constraints in Theorem 3.2, assume $L(t)$ enjoys linear convergence with rate $\bar{\rho}(\eta_0, 0)$ until iteration $k$, then the following holds for iteration $k+1$*

$$\mu_{k+1} \geq \bar{\mu}\,, \quad K_{k+1} \leq \bar{K}_{k+1}\,, \tag{23}$$

*with $\bar{\mu}, \bar{K}_{k+1}$ defined in Theorem 3.2. Moreover, one can show*

$$\rho(\eta_{k+1}, k+1) \leq \bar{\rho}(\eta_{k+1}, k+1) \leq \bar{\rho}(\eta_0, 0)\,. \tag{24}$$

Equation 23 is a direct consequence of Lemma 3.1 and the induction that $L(t)$ enjoys linear convergence until iteration $k$. We can bound $\mu_k, K_k$ by subsituting $\sigma_{\min}(\mathcal{T}_k), \sigma_{\max}(\mathcal{T}_k), \sigma_{\max}(W(k)), L(k)$ with the bounds in equation 21, equation 22 and $L(0)\bar{\rho}(\eta_0, 0)^t$ respectively. Based on these results, one can derive the following upper bound on $L(k+1)$ under the same constraints on $\eta_t$ in Theorem 3.2

$$L(k+1) \leq (1 - 2\bar{\mu}\eta_k + \bar{\mu}K_k\eta_k^2)L(k) \leq (1 - 2\bar{\mu}\eta_k + \bar{\mu}\bar{K}_0\eta_k^2)L(k) \leq \bar{\rho}(\eta_0, 0)L(k)\,, \tag{25}$$

where the first inequality is based on Descent lemma and PL inequality with constant $\bar{\mu}$ in Theorem 3.2. The second inequality is based on the fact that $K_k \leq \bar{K}_k \leq \bar{K}_0$, and the third one is derived using constraints on $\eta_0$ and equation 15.

## 4 CONCLUSION

This paper studied the convergence of GD for optimizing two-layer linear networks on general loss. In particular, we derived a convergence rate for networks of finite width that are initialized in a non-NTK regime. We use a common framework for studying the convergence of GD for the non-convex optimization problem, i.e. PL condition and Descent lemma. Although the loss landscape of neural networks does not satisfy PL condition and Descent lemma with global constants, we show that when the step size is small, both conditions satisfy locally with constants depending on the singular value of the weights, the current loss, and the singular value of the products. Furthermore, We prove that the local PL constants and smoothness constants can be bounded uniformly by the initial imbalance, the margin, the PL constant, and the smoothness constant of the non-overparametrized models. In addition, we derive an explicit convergence rate that depends on the margin, imbalance, and condition number of the non-overparametrized model. Finally, based on the convergence rate, we propose an adaptive step size scheme that accelerates convergence compared with a constant step size. Empirically, we show the convergence rate derived in our work is tighter than in previous work.

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

## A PRELIMINARY LEMMAS

In this section, we present a preliminary lemma which will be used in the following sections.

**Lemma A.1** (Inequality on the Frobenius norm). *For matrix $A, B, C, D$, we have*

$$\langle A, B \rangle \leq \|A\|_F \cdot \|B\|_F , \tag{26}$$
$$2\|AB\|_F \leq \|A\|_F^2 + \|B\|_F^2 , \tag{27}$$
$$\|AB + CD\|_F^2 \leq [\sigma_{\max}^2(A) + \sigma_{\max}^2(C)]^2 \cdot [\|B\|_F^2 + \|D\|_F^2] , \tag{28}$$
$$\|A\|_F^2 + \|B\|_F^2 \leq 2\|A + B\|_F^2 , \tag{29}$$
$$\sigma_{\min}^2(A)\|B\|_F^2 \leq \|AB\|_F^2 \leq \sigma_{\max}^2(A)\|B\|_F^2 , \tag{30}$$
$$\sigma_{\min}^2(B)\|A\|_F^2 \leq \|AB\|_F^2 \leq \sigma_{\max}^2(B)\|A\|_F^2 . \tag{31}$$

Lemma A.1 has been derived and used multiple times in prior work. We refer the readers to Appendix C in Xu et al. (2023) for detailed proof.

**Lemma A.2** (Singular values of $\mathcal{T}$). *The largest and smallest singular values of $\mathcal{T}$ are given as*

$$\sigma_{\min}^2(\mathcal{T}) = \sigma_{\min}^2(W_1) + \sigma_{\min}^2(W_2) ,$$
$$\sigma_{\max}^2(\mathcal{T}) = \sigma_{\max}^2(W_1) + \sigma_{\max}^2(W_2) . \tag{32}$$

*Proof.* First, one can see

$$\mathcal{T}^* \circ \mathcal{T}(U; W_1, W_2) = UW_2W_2^\top + W_1W_1^\top U , \tag{33}$$

where $\mathcal{T}^*$ is the adjoint of $\mathcal{T}$. Then, we use Min-max theorem to show

$$\lambda_{\min}(\mathcal{T}^* \circ \mathcal{T}) = \sigma_{\min}^2(W_1) + \sigma_{\min}^2(W_2), \quad \lambda_{\max}(\mathcal{T}^* \circ \mathcal{T}) = \sigma_{\max}^2(W_1) + \sigma_{\max}^2(W_2). \tag{34}$$

Let the singular value decompositions of $W_1, W_2$ be

$$W_1 = U_1\Sigma_1V_1^\top = \sum_{i=1}^{r_1} \sigma_{1,i}u_{1,i}v_{1,i}^\top , \quad W_2 = U_2\Sigma_2V_2^\top = \sum_{i=1}^{r_2} \sigma_{2,i}u_{2,i}v_{2,i}^\top , \tag{35}$$

where $r_1 = \text{rank}(W_1), r_2 = \text{rank}(W_2)$, and $\{\sigma_{1,i}\}_{i=1}^{r_1}, \{\sigma_{2,i}\}_{i=1}^{r_2}$ are of descending order. Then, one has the following

$$\begin{aligned}
\lambda_{\min}(\mathcal{T}^* \circ \mathcal{T}) &= \min_{\|U\|_F=1} \langle U, UW_2W_2^\top + W_1W_1^\top U \rangle \\
&= \min_{\|U\|_F=1} \langle U, UW_2W_2^\top \rangle + \min_{\|U\|_F=1} \langle U, W_1W_1^\top U \rangle \\
&\geq \sigma_{\min}^2(W_1) + \sigma_{\min}^2(W_2) .
\end{aligned} \tag{36}$$

