Justin Sirignano and Konstantinos Spiliopoulos. Mean field analysis of neural networks: A law of large numbers. *SIAM Journal on Applied Mathematics*, 80(2):725–752, 2020.

Samuel L Smith, Benoit Dherin, David GT Barrett, and Soham De. On the origin of implicit regularization in stochastic gradient descent. *arXiv preprint arXiv:2101.12176*, 2021.

Ruoyu Sun, Dawei Li, Shiyu Liang, Tian Ding, and Rayadurgam Srikant. The global landscape of neural networks: An overview. *IEEE Signal Processing Magazine*, 37(5):95–108, 2020.

Salma Tarmoun, Guilherme Franca, Benjamin D Haeffele, and Rene Vidal. Understanding the dynamics of gradient flow in overparameterized linear models. In *International Conference on Machine Learning*, pp. 10153–10161. PMLR, 2021.

Ashish Vaswani, Noam Shazeer, Niki Parmar, Jakob Uszkoreit, Llion Jones, Aidan N Gomez, Ł ukasz Kaiser, and Illia Polosukhin. Attention is all you need. In I. Guyon, U. Von Luxburg, S. Bengio, H. Wallach, R. Fergus, S. Vishwanathan, and R. Garnett (eds.), *Advances in Neural Information Processing Systems*, volume 30. Curran Associates, Inc., 2017. URL https://proceedings.neurips.cc/paper_files/paper/2017/file/3f5ee243547dee91fbd053c1c4a845aa-Paper.pdf.

Roman Vershynin. *High-dimensional probability: An introduction with applications in data science*, volume 47. Cambridge university press, 2018.

Mingwei Wei and David J Schwab. How noise affects the hessian spectrum in overparameterized neural networks. *arXiv preprint arXiv:1910.00195*, 2019.

Yikai Wu, Xingyu Zhu, Chenwei Wu, Annie Wang, and Rong Ge. Dissecting hessian: Understanding common structure of hessian in neural networks, 2022.

Ziqing Xu, Hancheng Min, Salma Tarmoun, Enrique Mallada, and René Vidal. Linear convergence of gradient descent for finite width over-parametrized linear networks with general initialization. In *International Conference on Artificial Intelligence and Statistics*, pp. 2262–2284. PMLR, 2023.

Jingzhao Zhang, Tianxing He, Suvrit Sra, and Ali Jadbabaie. Why gradient clipping accelerates training: A theoretical justification for adaptivity, 2020.

Yi Zhou and Yingbin Liang. Characterization of gradient dominance and regularity conditions for neural networks. *arXiv preprint arXiv:1710.06910*, 2017.

## A    PRELIMINARY LEMMAS

In this section, we present a preliminary lemma which will be used in the following sections.

**Lemma A.1** (Inequality on the Frobenius norm). *For matrix $A, B, C, D$, we have*

$$\langle A, B \rangle \leq \|A\|_F \cdot \|B\|_F \,, \tag{26}$$

$$2\|AB\|_F \leq \|A\|_F^2 + \|B\|_F^2 \,, \tag{27}$$

$$\|AB + CD\|_F^2 \leq [\sigma_{\max}^2(A) + \sigma_{\max}^2(C)]^2 \cdot [\|B\|_F^2 + \|D\|_F^2] \,, \tag{28}$$

$$\|A\|_F^2 + \|B\|_F^2 \leq 2\|A + B\|_F^2 \,, \tag{29}$$

$$\sigma_{\min}^2(A)\|B\|_F^2 \leq \|AB\|_F^2 \leq \sigma_{\max}^2(A)\|B\|_F^2 \,, \tag{30}$$

$$\sigma_{\min}^2(B)\|A\|_F^2 \leq \|AB\|_F^2 \leq \sigma_{\max}^2(B)\|A\|_F^2 \,. \tag{31}$$

Lemma A.1 has been derived and used multiple times in prior work. We refer the readers to Appendix C in Xu et al. (2023) for detailed proof.

**Lemma A.2** (Singular values of $\mathcal{T}$). *The largest and smallest singular values of $\mathcal{T}$ are given as*

$$\sigma_{\min}^2(\mathcal{T}) = \sigma_{\min}^2(W_1) + \sigma_{\min}^2(W_2) \,,$$
$$\sigma_{\max}^2(\mathcal{T}) = \sigma_{\max}^2(W_1) + \sigma_{\max}^2(W_2) \,. \tag{32}$$

*Proof.* First, one can see

$$\mathcal{T}^* \circ \mathcal{T}(U; W_1, W_2) = U W_2 W_2^\top + W_1 W_1^\top U, \tag{33}$$

where $\mathcal{T}^*$ is the adjoint of $\mathcal{T}$. Then, we use Min-max theorem to show

$$\lambda_{\min}(\mathcal{T}^* \circ \mathcal{T}) = \sigma_{\min}^2(W_1) + \sigma_{\min}^2(W_2), \quad \lambda_{\max}(\mathcal{T}^* \circ \mathcal{T}) = \sigma_{\max}^2(W_1) + \sigma_{\max}^2(W_2). \tag{34}$$

Let the singular value decompositions of $W_1, W_2$ be

$$W_1 = U_1 \Sigma_1 V_1^\top = \sum_{i=1}^{r_1} \sigma_{1,i} u_{1,i} v_{1,i}^\top, \quad W_2 = U_2 \Sigma_2 V_2^\top = \sum_{i=1}^{r_2} \sigma_{2,i} u_{2,i} v_{2,i}^\top, \tag{35}$$

where $r_1 = \mathrm{rank}(W_1), r_2 = \mathrm{rank}(W_2)$, and $\{\sigma_{1,i}\}_{i=1}^{r_1}, \{\sigma_{2,i}\}_{i=1}^{r_2}$ are of descending order. Then, one has the following

$$\begin{aligned}
\lambda_{\min}(\mathcal{T}^* \circ \mathcal{T}) &= \min_{\|U\|_F=1} \langle U, U W_2 W_2^\top + W_1 W_1^\top U \rangle \\
&= \min_{\|U\|_F=1} \langle U, U W_2 W_2^\top \rangle + \min_{\|U\|_F=1} \langle U, W_1 W_1^\top U \rangle \\
&\geq \sigma_{\min}^2(W_1) + \sigma_{\min}^2(W_2).
\end{aligned} \tag{36}$$

On the other hand, if one choose $U = v_{1,r_1} u_{2,r_2}^\top$, the following equation holds

$$\begin{aligned}
&\langle U, U W_2 W_2^\top + W_1 W_1^\top U \rangle \\
=& \langle v_{1,r_1} u_{2,r_2}^\top, v_{1,r_1} u_{2,r_2}^\top W_2 W_2^\top + W_1 W_1^\top v_{1,r_1} u_{2,r_2}^\top \rangle \\
=& \langle v_{1,r_1} u_{2,r_2}^\top, v_{1,r_1} u_{2,r_2}^\top \sum_{i=1}^{r_2} \sigma_{2,i}^2 u_{2,i} v_{2,i}^\top \rangle + \langle v_{1,r_1} u_{2,r_2}^\top, \sum_{i=1}^{r_1} \sigma_{1,i}^2 u_{1,i} v_{1,i}^\top v_{1,r_1} u_{2,r_2}^\top \rangle \\
=& \sum_{i=1}^{r_2} \sigma_{2,i}^2 \mathrm{tr}(u_{2,r_2} v_{1,r_1}^\top v_{1,r_1} u_{2,r_2}^\top u_{2,i} v_{2,i}^\top) + \sum_{i=1}^{r_2} \sigma_{1,i}^2 \mathrm{tr}(u_{2,r_2} v_{1,r_1}^\top u_{1,i} v_{1,i}^\top v_{1,r_1} u_{2,r_2}^\top) \\
=& \sigma_{1,r_1}^2 + \sigma_{2,r_2}^2,
\end{aligned} \tag{37}$$

where the last line is based on the fact that $v_{1,i}^\top v_{i,r_1} = 0, u_{2,j}^\top u_{2,r_2} = 0$ holds for all $i \neq r_1, j \neq r_2$. Therefore, based on equation 36 and equation 37, one has

$$\lambda_{\min}(\mathcal{T}^* \circ \mathcal{T}) = \sigma_{\min}^2(W_1) + \sigma_{\min}^2(W_2). \tag{38}$$

Similarly, we can show

$$\lambda_{\max}(\mathcal{T}^* \circ \mathcal{T}) = \sigma_{\max}^2(W_1) + \sigma_{\max}^2(W_2). \tag{39}$$

$\square$

## B  NON-EXISTENCE OF GLOBAL PL CONSTANT AND SMOOTHNESS CONSTANT FOR PROBLEM 2

In this section, we show that under mild assumptions, the PL inequality and smoothness inequality can only hold with constants $\mu_{\mathrm{over}} = 0$ and $K_{\mathrm{over}} = \infty$ for Problem 2.

Before presenting the results, we first present the following **preliminary lemma** on the inequality of the Frobenius norm.

We then make the following assumption on Problem 2.

**Assumption B.1.** $(W_1, W_2) = (0, 0)$ *is not a global minimizer of Problem* 2.

Based on the above assumption, one has the following proposition.

**Proposition B.1** (Non-existence of global PL constant and smoothness constant). *Under Assumption B.1, the PL inequality and smoothness inequality can only hold with constants $\mu_{over} = 0$ and $K_{over} = \infty$ for $L(W_1, W_2)$.*

*Proof.* We first show $\mu_{\text{over}} = 0$. The gradient of $L$ is given as follows

$$\|\nabla L(W_1, W_2)\|_F^2 = \|\nabla \ell(W) W_2\|_F^2 + \|\nabla \ell(W)^\top W_1\|_F^2. \tag{40}$$

Notice when $W_1, W_2$ are zero matrices, the RHS of the above equation is zero. Therefore, we have $\|\nabla L(W_1, W_2)\|_F^2 = 0$. On the other hand, under Assumption B.1, since $(W_1, W_2) = (0,0)$ is not a minimizer of Problem 2, we have $L(W_1, W_2) \neq 0$. Thus, the PL inequality can only hold globally with $\mu_{\text{over}} = 0$,

$$\|\nabla L(W_1, W_2)\|_F^2 = \|\nabla \ell(W) W_2\|_F^2 + \|\nabla \ell(W)^\top W_1\|_F^2 \geq 2\mu_{\text{over}} L(W_1, W_2). \tag{41}$$

Then, we show $K_{\text{over}} = \infty$ for Problem 2. We consider the smoothness inequality evaluated on arbitrary $(W_1, W_2)$ and the minimizer $(W_1^*, W_2^*)$ of Problem 2:

$$L(W_1, W_2) \leq L(W_1^*, W_2^*) + \langle \nabla L(W_1^*, W_2^*), Z^* - Z \rangle + \frac{K_{\text{over}}}{2} \|Z^* - Z\|_F^2, \tag{42}$$

where we use $Z, Z^*$ in short for $(W_1, W_2), (W_1^*, W_2^*)$. Since $(W_1^*, W_2^*)$ minimizes Problem 2, we have $\nabla L(W_1^*, W_2^*) = 0_{(m+n) \times h}$ and $L(W_1^*, W_2^*) = 0$. Thus, equation 42 is equivalent to the following

$$K_{\text{over}} \geq \frac{2L(W_1, W_2)}{\|Z_W - Z_{W^*}\|_F^2} = \frac{2\ell(W_1 W_2^\top)}{\|Z_W - Z_{W^*}\|_F^2}. \tag{43}$$

On the other hand, since $\ell(W)$ is $\mu$-strongly convex w.r.t. $W$, the following inequality holds for arbitrary $U, V$

$$\ell(U) \geq \ell(V) + \langle \nabla \ell(V), U - V \rangle + \frac{\mu}{2} \|U - V\|_F^2. \tag{44}$$

We substitute $U, V$ with $W_1 W_2^\top, W_1^*(W_2^*)^\top$ in equation 44, we have

$$\ell(W_1 W_2^\top) \geq \frac{\mu}{2} \|W_1 W_2^\top - W_1^*(W_2^*)^\top\|_F^2. \tag{45}$$

Finally, we combine equation 43 and equation 45, and derive the following lower bound on $K_{\text{over}}$

$$
\begin{aligned}
K_{\text{over}} &\geq \frac{2\ell(W_1 W_2^\top)}{\|Z_W - Z_{W^*}\|_F^2} && \text{Based on equation 43} \\[2mm]
&\geq \frac{\mu \|W_1 W_2^\top - W_1^*(W_2^*)^\top\|_F^2}{\|W_1 - W_1^*\|_F^2 + \|W_2 - W_2^*\|_F^2} && \text{Apply equation 45 to } \ell(W_1 W_2^*) \\[2mm]
&= \frac{\mu \|W_1 W_2^\top - W_1^* W_2^\top + W_1^* W_2^\top - W_1^*(W_2^*)^\top\|_F^2}{\|W_1 - W_1^*\|_F^2 + \|W_2 - W_2^*\|_F^2} \\[2mm]
&= \frac{\mu \|(W_1 - W_1^*) W_2^\top + W_1^*(W_2 - W_2^*)^\top)\|_F^2}{\|W_1 - W_1^*\|_F^2 + \|W_2 - W_2^*\|_F^2} \\[2mm]
&\geq \frac{\mu}{2} \cdot \frac{\|(W_1 - W_1^*) W_2^\top\|_F^2 + \|W_1^*(W_2 - W_2^*)^\top)\|_F^2}{2\|W_1 - W_1^*\|_F^2 + \|W_2 - W_2^*\|_F^2} && \text{Apply Lemma A.1} \\[2mm]
&\geq \frac{\mu}{2} \cdot \frac{\sigma_{\min}^2(W_2)\|W_1 - W_1^*\|_F^2 + \sigma_{\min}^2(W_1^*)\|W_2 - W_2^*\|_F^2}{\|W_1 - W_1^*\|_F^2 + \|W_2 - W_2^*\|_F^2} && \text{Apply Lemma A.1.} \quad (46)
\end{aligned}
$$

Similarly, one can also derive the following lower bound on $K_{\text{over}}$

$$K_{\text{over}} \geq \frac{\mu}{2} \cdot \frac{\sigma_{\min}^2(W_2^*)\|W_1 - W_1^*\|_F^2 + \sigma_{\min}^2(W_1)\|W_2 - W_2^*\|_F^2}{\|W_1 - W_1^*\|_F^2 + \|W_2 - W_2^*\|_F^2} \tag{47}$$

We take the average of the lower bound on $K_{\text{over}}$ in equation 46 and equation 47,

$$
\begin{aligned}
K_{\text{over}} &\geq \frac{\mu}{4} \cdot \frac{\left(\sigma_{\min}^2(W_2) + \sigma_{\min}^2(W_2^*)\right)\|W_1 - W_1^*\|_F^2 + \left(\sigma_{\min}^2(W_1) + \sigma_{\min}^2(W_1^*)\right)\|W_2 - W_2^*\|_F^2}{\|W_1 - W_1^*\|_F^2 + \|W_2 - W_2^*\|_F^2} \\[2mm]
&\geq \frac{\mu}{4} \cdot \min\left(\sigma_{\min}^2(W_1) + \sigma_{\min}^2(W_1^*), \sigma_{\min}^2(W_2) + \sigma_{\min}^2(W_2^*)\right) \\[2mm]
&\geq \frac{\mu}{4} \cdot \min\left(\sigma_{\min}^2(W_1), \sigma_{\min}^2(W_2)\right). \tag{48}
\end{aligned}
$$

Due to the arbitrary choices of $W_1, W_2$, we can let $\sigma_{\min}(W_1)$ and $\sigma_{\min}(W_2)$ to be arbitrarily large, thus the smoothness inequality for Problem 2 can only hold globally with $K_{\text{over}} = \infty$. $\qquad\square$

## C  A DETAILED COMPARISON WITH PRIOR WORK

In this section, we present a detailed comparison to Arora et al. (2018); Du et al. (2018a); Xu et al. (2023) to highlight the difference in technical details and improvement on the convergence rate.

**Summary of the strategy of proof in (Arora et al., 2018; Du et al., 2018a; Xu et al., 2023).** Based on GD update in equation 8, one can derive the following update on the product $W(t)$

$$W(t+1) = W(t) - \eta_t \mathcal{T}_t^* \circ \mathcal{T}_t(\nabla\ell(t)) + \eta_t^2 \nabla\ell(t)W(t)^\top \nabla\ell(t) , \tag{49}$$

where $\mathcal{T}_t^*$ is the adjoint of $\mathcal{T}_t$. Then, substituting equation 49 into the smoothness inequality of the non-overparametrized model in equation 4, we can derive the following upper bound on the loss at iteration $t+1$ using the loss at iteration $t$ (Also see Lemma3.1 in Xu et al. (2023)).

**Lemma C.1.** *If at the $t$-th iteration of GD applied to the over-parametrized loss $L$, the step size $\eta_t$ satisfies*

$$\sigma_{\min}^2(\mathcal{T}_t) - \eta_t\|\nabla\ell(t)\|_F\|W(t)\|_F - \frac{K\eta_t}{2}\left[\sigma_{\max}^2(\mathcal{T}_t) + \eta_t\|\nabla\ell(t)\|_F\|W(t)\|_F\right]^2 \geq 0 , \tag{50}$$

*then the following inequality holds*

$$L(t+1) \leq \rho(\eta,t)L(t) , \tag{51}$$

*where*

$$\rho(\eta,t) = 1 - 2\eta_t\mu\sigma_{\min}^2(\mathcal{T}_t) + K\mu\eta_t^2\sigma_{\max}^4(\mathcal{T}_t) + 2\eta_t^2\mu\sigma_{\max}(W(t))\|\nabla\ell(t)\|_F$$
$$+ 2\eta_t^3\mu K\sigma_{\max}^2(\mathcal{T}_t)\sigma_{\max}(W(t))\|\nabla\ell(t)\|_F + \eta_t^4\mu K\sigma_{\max}^2(W(t))\|\nabla\ell(t)\|_F^2. \tag{52}$$

**Improvement of the local rate of decrease.** First, one can see the local rates of decrease in both work are polynomials of degree four and depend on $\eta_t, \nabla\ell(t)$ and singular values of $\mathcal{T}_t, W_t$. Moreover, around any global minimum, i.e., $L(t) \approx 0, \|\nabla\ell(t)\|_F \approx 0$, we have the following local rate of decrease per iteration

$$1 - 2\eta_t\mu\sigma_{\min}^2(\mathcal{T}_t) + \eta_t^2 K\mu\sigma_{\max}^4(\mathcal{T}_t) \qquad \text{local rate of decrease in prior work} ,$$
$$1 - 2\eta_t\mu\sigma_{\min}^2(\mathcal{T}_t) + \eta_t^2 K\mu\sigma_{\min}^2(\mathcal{T}_t)\eta_t^2\sigma_{\max}^2(\mathcal{T}_t) \qquad \text{local rate of decrease in this work} , \tag{53}$$

and the optimal local rates of decrease regardless of the constraints on the $\eta_t$ are

$$1 - \frac{\mu}{K} \cdot \frac{\sigma_{\min}^4(\mathcal{T}_t)}{\sigma_{\max}^4(\mathcal{T}_t)} \qquad \text{optimal local rate of decrease in prior work} ,$$

$$1 - \frac{\mu}{K} \cdot \frac{\sigma_{\min}^2(\mathcal{T}_t)}{\sigma_{\max}^2(\mathcal{T}_t)} \qquad \text{optimal local rate of decrease in this work} . \tag{54}$$

Thus, one can see our characterization of local Descent lemma and PL inequality leads to faster local rates of decrease compared with prior results by $\frac{\sigma_{\min}^2(\mathcal{T}_t)}{\sigma_{\max}^2(\mathcal{T}_t)}$. Nevertheless, equation 54 does not imply linear convergence since if $\lim_{t\to\infty} \frac{\sigma_{\min}^2(\mathcal{T}_t)}{\sigma_{\max}^2(\mathcal{T}_t)} = 0$, one would not expect sufficient decrease per iteration. In order to show linear convergence, one needs to provide a uniform lower bound on $\frac{\sigma_{\min}^2(\mathcal{T}_t)}{\sigma_{\max}^2(\mathcal{T}_t)}, \forall t$.

**Improvement of the local rate of convergence.** In this work, we show when the step sizes satisfy certain constraints (See Theorem 3.2), there exist uniform spectral bounds for the condition number of $\mathcal{T}_t$, i.e., $\frac{\sigma_{\min}^2(\mathcal{T}_t)}{\sigma_{\max}^2(\mathcal{T}_t)} \leq c(\eta_0)\frac{\alpha_1}{\alpha_2}, \forall t$ where $\alpha_1, \alpha_2$ only depend on the initial weights and $c(\eta_0)$ is a constant approaching one as $\eta_0$ decreases. Thus, the optimal final rate of convergence derived in this work is

$$1 - \frac{\mu}{K} \cdot \frac{\alpha_1}{\alpha_2} \qquad \text{optimal local rate of convergence in this work} . \tag{55}$$

In prior work, the rates in (Du et al., 2018a; Arora et al., 2018) are extremely slow in practice (See Section 4 in (Xu et al., 2023)). In (Xu et al., 2023), the authors introduce two auxiliary constants $0 < c_1 < 1, c_2 > 1$, and show that one can uniformly bound the condition number of $\mathcal{T}_t$ during training,

i.e., $\frac{\sigma_{\min}^2(\mathcal{T}_t)}{\sigma_{\max}^2(\mathcal{T}_t)} \le \frac{c_1\alpha_1}{c_2\alpha_2}, \forall t$. Moreover, they enforce problem-dependent assumptions on the choices of $c_1, c_2$. According to Claim E.1 in (Xu et al., 2023), $\frac{c_1}{c_2}$ is at most $\frac{1}{3}$ and can be arbitrarily small when the initial loss is large. Thus, the local rate of convergence in (Xu et al., 2023) is at most, in our notation,

$$1 - \frac{\mu}{K} \cdot \frac{c_1^2\alpha_1^2}{c_2^2\alpha_2^2} \qquad \text{optimal local rate of convergence in (Xu et al., 2023)}. \tag{56}$$

When comparing equation 55 and equation 56, one can directly conclude the local rate of convergence derived in this work is much faster than the rate derived in (Xu et al., 2023). Moreover, the optimal local rate of convergence of the overparametrized model in this work is different from the optimal rate of convergence of the non-overparametrized model up to a factor of $\frac{\alpha_1}{\alpha_2}$, which shows overparametrization has a benign effect if one can control $\frac{\alpha_1}{\alpha_2}$ through properly initialization of the weights. However, such results are not shown in the work of (Arora et al., 2018; Du et al., 2018a; Xu et al., 2023).

## D  PROOF OF THEOREM 3.1

In this section, we present the proof of Theorem 3.1.

**Theorem D.1** (Restate of Theorem 3.1). *At the $t$-th iteration of GD applied to the Problem 2, the Descent lemma and PL inequality hold with local smoothness constant $K_t$ and PL constant $\mu_t$, i.e.,*

$$L(t+1) \le L(t) - \left(\eta_t - \frac{K_t\eta_t^2}{2}\right)\|\nabla L(t)\|_F^2, \quad \frac{1}{2}\|\nabla L(t)\|_F^2 \ge \mu_t L(t). \tag{57}$$

*Moreover, if the step size $\eta_t$ satisfies $\eta_t > 0$ and $\eta_t K_t < 2$, then the following inequality holds*

$$L(t+1) \le L(t)(1 - 2\mu_t\eta_t + \mu_t K_t\eta_t^2) := L(t)\rho(\eta_t, t), \tag{58}$$

*where*

$$\mu_t = \mu\sigma_{\min}^2(\mathcal{T}_t), \tag{59}$$

$$K_t = K\sigma_{\max}^2(\mathcal{T}_t) + \sqrt{2KL(t)} + 6K^2\sigma_{\max}(W(t))L(t)\eta_t^2 + 3K\sigma_{\max}^2(\mathcal{T}_t)\sqrt{2KL(t)}\eta_t, \tag{60}$$

*and we use $L(t), \mathcal{T}_t$ as a shorthand for $L(W_1(t), W_2(t)), \mathcal{T}(\,\cdot\,; W_1(t), W_2(t))$ resp.*

*Proof.* We first show that the local PL inequality holds.

$$\begin{aligned}
\|\nabla L(t)\|_F^2 &= \left\|\begin{bmatrix} \nabla\ell(t)W_2(t) \\ \nabla\ell(t)^\top W_1(t) \end{bmatrix}\right\|_F^2 \\
&= \|\ell(t)W_2(t)\|_F^2 + \|\ell(t)^\top W_1(t)\|_F^2 \\
&\ge \sigma_{\min}^2(W_2(t))\|\nabla\ell(t)\|_F^2 + \sigma_{\min}^2(W_1(t))\|\nabla\ell(t)\|_F^2 \qquad \text{Apply Lemma A.1} \\
&\ge 2\mu\sigma_{\min}^2(W_2(t))L(t) + 2\mu\sigma_{\min}^2(W_1(t))L(t) \qquad \text{Apply PL inequality of } \ell \\
&= 2\mu_t L(t),
\end{aligned}$$

where the last equality uses the fact that $\sigma_{\min}^2(\mathcal{T}_t) = \sigma_{\min}^2(W_1(t)) + \sigma_{\min}^2(W_2(t))$. Then, we show that Descent lemma holds with local smoothness constant $K_t$. We can view $L(t+1)$ using the following second Taylor approximation,

$$L(t+1) = L(t) + \langle\nabla L(t), Z_{t+1} - Z_t\rangle + \int_0^1 (1-\tau)\langle Z_{t+1} - Z_t, H(\tau)(Z_{t+1} - Z_t)\rangle d\tau,$$

$$= L(t) - \eta_t\|\nabla L(t)\|_F^2 + \eta_t^2\|\nabla L(t)\|_F^2 \int_0^1 (1-\tau)\langle g_t, H(\tau)g_t\rangle d\tau, \tag{61}$$

where we use $Z_{t+1}, Z_t$ in short for $(W_1(t+1), W_2(t+1)), (W_1(t), W_2(t))$ respectively, and $g_t = \frac{\nabla L(t)}{\|\nabla L(t)\|_F}$ to denote the unit vector of the gradient direction. Moreover, the $H(\tau)$ is defined as follows,

$$\begin{aligned}
H(\tau) &= \nabla^2 L\big((1-\tau)W_1(t) + \tau W_1(t+1), (1-\tau)W_2(t) + \tau W_2(t+1)\big) \\
&= \nabla^2 L\big(W_1(t) - \eta_t\tau\nabla_{W_1}L(t), W_2(t) - \eta_t\tau\nabla_{W_2}L(t)\big). \tag{62}
\end{aligned}$$

Notice the integral in the equation 61 does not have a closed-form solution. We use the following two-step approach to derive an upper bound on the RHS of equation 61.

**Step one.** We first show that one can upper bound $\langle g_t, H(0)g_t \rangle$ using the singular values of $\mathcal{T}_t$, $K$ and $L(t)$. The following lemma characterizes it formally.

**Lemma D.1** (Upper bound on $\langle g_t, H(0)g_t \rangle$). *We have the following upper bound*

$$\langle g_t, H(0)g_t \rangle \leq K\sigma_{\max}^2(\mathcal{T}_t) + \sqrt{2KL(t)}. \tag{63}$$

The proof of Lemma D.1 is presented at the end of this section.

**Step two.** Then, for any $\tau \in [0, 1)$, we can show $|\langle g_t, \big(H(0) - H(\tau)\big)g_t \rangle|$ is bounded, which leads to an upper bound on $\langle g_t, H(\tau)g_t \rangle$. The following lemma characterizes the upper bound on $\langle g_t, H(\tau)g_t \rangle$.

**Lemma D.2** (Uniform upper bound on $\langle g_t, H(\tau)g_t \rangle$). *For any $\tau \in [0, 1)$, we have*

$$|\langle g_t, H(\tau)g_t \rangle| \leq K\sigma_{\max}^2(\mathcal{T}_t) + \sqrt{2KL(t)} + 6K^2\sigma_{\max}(W(t))L(t)\eta_t^2 + 3K\sigma_{\max}^2(\mathcal{T}_t)\sqrt{2KL(t)}\eta_t := K_t.$$

The proof of Lemma D.2 is presented at the end of this section.

Based on equation 61 and Lemma D.2, one can derive Descent lemma

$$
\begin{aligned}
L(t+1) &= L(t) - \eta_t\|\nabla L(t)\|_F^2 + \eta_t^2\|\nabla L(t)\|_F^2 \int_0^1 (1-\tau)\langle g_t, H(\tau)g_t \rangle d\tau && \text{Equation 61} \\
&\leq L(t) - \eta_t\|\nabla L(t)\|_F^2 + \eta_t^2\|\nabla L(t)\|_F^2 \int_0^1 (1-\tau)\max_\tau |\langle g_t, H(\tau)g_t \rangle| d\tau \\
&\leq L(t) - \eta_t\|\nabla L(t)\|_F^2 + \eta_t^2\|\nabla L(t)\|_F^2 \int_0^1 (1-\tau)K_t d\tau && \text{Lemma D.2} \\
&= L(t) - \eta_t\|\nabla L(t)\|_F^2 + \frac{\eta_t^2 K_t}{2}\|\nabla L(t)\|_F^2 \\
&= L(t) - (\eta_t - \frac{\eta_t^2 K_t}{2})\|\nabla L(t)\|_F^2. && (64)
\end{aligned}
$$

Therefore, Descent lemma is proved. $\qquad\square$

Now we present the proof of Lemma D.1 and Lemma D.2. We first define the following quantity which will be used in the proof

$$M(s) = L(W_1(t) - s\eta_t\nabla_{W_1}L(t), W_2(t) - s\eta_t\nabla_{W_2}L(t)), \tag{65}$$

$$A(s) = W(t) - s\eta_t\big(\nabla_{W_2}L(t)W_2(t)^\top + W_1(t)\nabla_{W_1}L(t)^\top\big) + s^2\eta_t^2\nabla_{W_1}L(t)\nabla_{W_2}L(t)^\top, \tag{66}$$

where $A(s)$ is the product of $W_1(t) - s\eta_t\nabla_{W_1}L(t)$ and $W_2(t) - s\eta_t\nabla_{W_2}L(t)$. Moreover, we have $M(0) = L(t)$, $M(\eta_t) = L(t+1)$ and $M(s) = \ell\big(A(s)\big)$.

Then, we present several lemmas that will be used in the proof of Lemma D.1 and Lemma D.2.

**Lemma D.3.** *Given $W_1(t) \in \mathbb{R}^{n \times h}, W_2(t) \in \mathbb{R}^{m \times h}$ at $t$-th iteration, the following holds*

$$2\|\nabla_{W_1}L(t)\nabla_{W_2}L(t)^\top\|_F \leq \|\nabla_{W_1}L(t)\|_F^2 + \|\nabla_{W_2}L(t)\|_F^2 \tag{67}$$

$$\|\nabla_{W_1}L(t)\nabla_{W_2}L(t)^\top\|_F \leq 2K\sigma_{\max}(W(t))L(t). \tag{68}$$

*Proof.* Based on Lemma A.1, one has $2\|AB\|_F \leq \|A\|_F^2 + \|B\|_F^2$. Thus, let $A = \nabla_{W_1}L(t), B = \nabla_{W_2}L(t)$, and we complete the proof of equation 67.

For equation 68, one has the following

$$
\begin{aligned}
\|\nabla_{W_1}L(t)\nabla_{W_2}L(t)^\top\|_F &= \|\nabla\ell(t)W(t)^\top\nabla\ell(t)^\top\|_F \\
&\leq \sigma_{\max}(W(t))\|\nabla\ell(t)\|_F^2 && \text{equation 30 in Lemma A.1} \\
&\leq 2K\sigma_{\max}(W(t))L(t) && K\text{-smooth of } \ell, && (69)
\end{aligned}
$$

which completes the proof. $\qquad\square$

**Lemma D.4.** *Given $W_1(t) \in \mathbb{R}^{n \times h}, W_2(t) \in \mathbb{R}^{m \times h}$ at $t$-th iteration, the following holds*

$$\|\nabla_{W_2} L(t) W_2(t)^\top + W_1(t) \nabla_{W_1} L(t)^\top\|_F \leq \sigma_{\max}^2(\mathcal{T}_t) \sqrt{2KL(t)}. \tag{70}$$

*Proof.* We prove this lemma using the results from Lemma A.1 and Lemma A.2.

$$
\begin{aligned}
& \|\nabla_{W_2} L(t) W_2(t)^\top + W_1(t) \nabla_{W_1} L(t)^\top\|_F \\
\leq & \|\nabla_{W_2} L(t) W_2(t)^\top\|_F + \|W_1(t) \nabla_{W_1} L(t)^\top\|_F && \text{Property of norm} \\
= & \|\nabla\ell(t) W_2 W_2(t)^\top\|_F + \|W_1(t) W_1(t)^\top \nabla\ell(t)^\top\|_F && \text{See definition of } \mathcal{T}_t \\
\leq & \sigma_{\max}^2(W_2(t)) \|\nabla\ell(t)\|_F^2 + \sigma_{\max}^2(W_1(t)) \|\nabla\ell(t)\|_F^2 && \text{equation 30 in Lemma A.1} \\
= & \sigma_{\max}^2(\mathcal{T}_t) \|\nabla\ell(t)\|_F^2 && \text{Lemma A.2} \\
\leq & \sigma_{\max}^2(\mathcal{T}_t) \sqrt{2KL(t)} && K\text{-smooth of } \ell. \tag{71}
\end{aligned}
$$

$\square$

**Lemma D.5.** *Given $W_1(t) \in \mathbb{R}^{n \times h}, W_2(t) \in \mathbb{R}^{m \times h}$ at $t$-th iteration, for any $s \in (0, 1]$, the following holds*

$$\|\nabla\ell\big(A(s)\big) - \nabla\ell\big(A(0)\|_F \leq \eta_t K \sqrt{2KL(t)} \sigma_{\max}^2(\mathcal{T}_t) + 2\eta_t^2 K^2 \sigma_{\max}(W(t)) L(t) \tag{72}$$

*Proof.* Based on Lemma A.1 and the assumption that $\ell$ is $K$-smooth, one has the following

$$
\begin{aligned}
& \|\nabla\ell\big(A(s)\big) - \nabla\ell\big(A(0)\|_F \\
\leq & K\|A(s) - A(0)\|_F \\
= & K\| - s\eta_t \big(\nabla_{W_2} L(t) W_2(t)^\top + W_1(t) \nabla_{W_1} L(t)^\top\big) + s^2 \eta_t^2 \nabla_{W_1} L(t) \nabla_{W_2} L(t)^\top\|_F \\
\leq & s\eta_t K\|\nabla_{W_2} L(t) W_2(t)^\top + W_1(t) \nabla_{W_1} L(t)^\top\|_F + s^2 \eta_t^2 K\|\nabla_{W_1} L(t) \nabla_{W_2} L(t)^\top\|_F \\
\leq & \eta_t K\|\nabla_{W_2} L(t) W_2(t)^\top + W_1(t) \nabla_{W_1} L(t)^\top\|_F + \eta_t^2 K\|\nabla_{W_1} L(t) \nabla_{W_2} L(t)^\top\|_F, \tag{73}
\end{aligned}
$$

where the last line is due to the fact that $s \in (0, 1]$.

Then, based on Lemma D.3 and Lemma D.4, one has the following,

$$
\begin{aligned}
& \|\nabla\ell\big(A(s)\big) - \nabla\ell\big(A(0)\|_F \\
\leq & K\|A(s) - A(0)\|_F \\
\leq & \eta_t K\|\nabla_{W_2} L(t) W_2(t)^\top + W_1(t) \nabla_{W_1} L(t)^\top\|_F + \eta_t^2 K\|\nabla_{W_1} L(t) \nabla_{W_2} L(t)^\top\|_F \\
\leq & \eta_t K \sigma_{\max}^2(\mathcal{T}_t) \sqrt{2KL(t)} + \eta_t^2 K \cdot 2K \sigma_{\max}(W(t)) L(t), \tag{74}
\end{aligned}
$$

which completes the proof. $\square$

**Lemma D.1** (Upper bound on $\langle g_t, H(0) g_t \rangle$)**.** *We have the following upper bound*

$$\langle g_t, H(0) g_t \rangle \leq K \sigma_{\max}^2(\mathcal{T}_t) + \sqrt{2KL(t)}. \tag{75}$$

*Proof.* First, we notice $\langle g_t, H(0) g_t \rangle$ is the second-order directional derivative of $L(t)$ w.r.t. the gradient direction,

$$\langle g_t, H(0) g_t \rangle = \frac{1}{\|\nabla L(t)\|_F^2} \cdot \frac{d^2}{ds^2} M(s) \bigg|_{s=0}. \tag{76}$$

Moreover, we can compute $\frac{d^2}{ds^2}M(s)\big|_{s=0}$ as follows

$$
\begin{aligned}
&\frac{d^2}{ds^2}M(s)\bigg|_{s=0}\\
&=\frac{d^2}{ds^2}L\bigg(\big(W_1(t)-s\nabla_{W_1}L(t)\big)\big(W_2(t)-s\nabla_{W_2}L(t)\big)^\top\bigg)\bigg|_{s=0}\\
&=\frac{d^2}{ds^2}\ell\big(A(s)\big)\bigg|_{s=0} && \text{Definition of } A(s)\\
&=\frac{d}{ds}\big\langle\nabla\ell\big(A(s)\big),\frac{d}{ds}A(s)\big\rangle\bigg|_{s=0}\\
&=\big\langle\nabla\ell\big(A(s)\big),\frac{d^2}{ds^2}A(s)\big\rangle+\big\langle\frac{d}{ds}A(s),\nabla^2\ell(A(s))\frac{d}{ds}A(s)\big\rangle\bigg|_{s=0}.
\end{aligned}
\tag{77}
$$

Under the assumption that $\ell$ is $K$-smooth and Lemma A.1, one can derive the following upper bound on $\frac{d^2}{ds^2}M(s)\big|_{s=0}$

$$
\begin{aligned}
&\frac{d^2}{ds^2}M(s)\bigg|_{s=0}\\
&=\big\langle\nabla\ell\big(A(s)\big),\frac{d^2}{ds^2}A(s)\big\rangle+\big\langle\frac{d}{ds}A(s),\nabla^2\ell(A(s))\frac{d}{ds}A(s)\big\rangle\bigg|_{s=0}\\
&\leq\big\langle\nabla\ell\big(A(s)\big),\frac{d^2}{ds^2}A(s)\big\rangle+K\|\frac{d}{ds}A(s)\|_F^2\bigg|_{s=0} && \ell \text{ is } K\text{-smooth}\\
&=2\langle\nabla\ell(t),\nabla_{W_1}L(t)\nabla_{W_2}L(t)^\top\rangle+K\|\nabla_{W_2}L(t)W_2(t)^\top+W_1(t)\nabla_{W_1}L(t)^\top\|_F^2\\
&\leq2\|\nabla\ell(t)\|_F\cdot\|\nabla_{W_1}L(t)\nabla_{W_2}L(t)^\top\|_F\\
&\quad+K[\sigma_{\max}^2(W_1(t))+\sigma_{\max}^2(W_2(t))]\cdot[\|\nabla_{W_1}L(t)\|_F^2+\|\nabla_{W_2}L(t)\|_F^2] && \text{Lemma A.1}\\
&\leq\|\nabla\ell(t)\|_F\cdot[\|\nabla_{W_1}L(t)\|_F^2+\|\nabla_{W_2}L(t)\|_F^2]\\
&\quad+K[\sigma_{\max}^2(W_1(t))+\sigma_{\max}^2(W_2(t))]\cdot[\|\nabla_{W_1}L(t)\|_F^2+\|\nabla_{W_2}L(t)\|_F^2] && \text{Lemma A.1}\\
&\leq\sqrt{2KL(t)}\cdot[\|\nabla_{W_1}L(t)\|_F^2+\|\nabla_{W_2}L(t)\|_F^2] && \text{K-smooth of } \ell\\
&\quad+K[\sigma_{\max}^2(W_1(t))+\sigma_{\max}^2(W_2(t))]\cdot[\|\nabla_{W_1}L(t)\|_F^2+\|\nabla_{W_2}L(t)\|_F^2].
\end{aligned}
\tag{78}
$$

Finally, we derive the the upper bound on $\langle g_t,H(0)g_t\rangle$ based on equation 78

$$
\begin{aligned}
\langle g_t,H(0)g_t\rangle&=\frac{1}{\|\nabla L(t)\|_F^2}\cdot\frac{d^2}{ds^2}M(s)\bigg|_{s=0}\\
&\leq\frac{[\|\nabla_{W_1}L(t)\|_F^2+\|\nabla_{W_2}L(t)\|_F^2]\cdot\big(\sqrt{2KL(t)}+\sigma_{\max}^2(W_1(t))+\sigma_{\max}^2(W_2(t))\big)}{\|\nabla_{W_1}L(t)\|_F^2+\|\nabla_{W_2}L(t)\|_F^2}\\
&=\sqrt{2KL(t)}+\sigma_{\max}^2(W_1(t))+\sigma_{\max}^2(W_2(t))\\
&=\sqrt{2KL(t)}+\sigma_{\max}^2(\mathcal{T}_t),
\end{aligned}
\tag{79}
$$

where the last line is based on Lemma A.2. $\qquad\square$

**Lemma D.2** (Upper bound on $\langle g_t,H(\tau)g_t\rangle$)**.** *For any* $\tau\in[0,1)$, *we have*

$$
\langle g_t,H(\tau)g_t\rangle\leq K_t,
\tag{80}
$$

*where*

$$
K_t=K\sigma_{\max}^2(\mathcal{T}_t)+\sqrt{2KL(t)}+6K^2\sigma_{\max}^2(W(t))L(t)\eta_t^2+3K\sigma_{\max}^2(\mathcal{T}_t)\sqrt{2KL(t)}\eta_t.
\tag{81}
$$

*Proof.* First, we use the same method to compute $\langle g_t,H(\tau)g_t\rangle$ as it was done in Lemma D.1

$$
\langle g_t,H(\tau)g_t\rangle=\frac{1}{\|\nabla L(t)\|_F^2}\cdot\frac{d^2}{ds^2}M(s+\tau)\bigg|_{s=0}.
\tag{82}
$$

Based on similar calculations in equation 77, one has

$$
\begin{aligned}
\frac{d^2}{ds^2} &M(s+\tau)\Big|_{s=0} \\
&= \frac{d^2}{ds^2} \ell\big(A(s+\tau)\big)\Big|_{s=0} \\
&= \frac{d}{ds}\big\langle \nabla\ell\big(A(s+\tau)\big), \frac{d}{ds}A(s+\tau)\big\rangle\Big|_{s=0} \\
&= \big\langle \nabla\ell\big(A(s+\tau)\big), \frac{d^2}{ds^2}A(s+\tau)\big\rangle + \big\langle \frac{d}{ds}A(s+\tau), \nabla^2\ell\big(A(s+\tau)\big)\frac{d}{ds}A(s+\tau)\big\rangle\Big|_{s=0}.
\end{aligned}
\tag{83}
$$

Under the assumption that $\ell$ is $K$-smooth, Lemma A.1 and Lemma D.1, one can show

$$
\begin{aligned}
\frac{d^2}{ds^2} &M(s+\tau)\Big|_{s=0} \\
&= \big\langle \nabla\ell\big(A(s+\tau)\big), \frac{d^2}{ds^2}A(s+\tau)\big\rangle + \big\langle \frac{d}{ds}A(s+\tau), \nabla^2\ell\big(A(s+\tau)\big)\frac{d}{ds}A(s+\tau)\big\rangle\Big|_{s=0} \\
&\leq \big\langle \nabla\ell\big(A(s+\tau)\big), \frac{d^2}{ds^2}A(s+\tau)\big\rangle + K\|\frac{d}{ds}A(s+\tau)\|_F^2\Big|_{s=0} \\
&= 2\big\langle \nabla\ell\big(A(\tau)\big), \nabla_{W_1}L(t)\nabla_{W_2}L(t)^\top\big\rangle \\
&\quad + K\|\nabla_{W_2}L(t)W_2(t)^\top + W_1(t)\nabla_{W_1}L(t)^\top - 2\tau\eta_t\nabla_{W_1}L(t)\nabla_{W_2}L(t)^\top\|_F^2 \\
&= 2\big\langle \nabla\ell\big(A(\tau)\big) - \nabla\ell\big(A(0)\big), \nabla_{W_1}L(t)\nabla_{W_2}L(t)^\top\big\rangle + 2\big\langle \nabla\ell\big(A(0)\big), \nabla_{W_1}L(t)\nabla_{W_2}L(t)^\top\big\rangle \\
&\quad + K\|\nabla_{W_2}L(t)W_2(t)^\top + W_1(t)\nabla_{W_1}L(t)^\top\|_F^2 + 4\tau^2\eta_t^2 K\|\nabla_{W_1}L(t)\nabla_{W_2}L(t)^\top\|_F^2 \\
&\quad - 4\tau K\eta_t\big\langle \nabla_{W_2}L(t)W_2(t)^\top + W_1(t)\nabla_{W_1}L(t)^\top, \nabla_{W_1}L(t)\nabla_{W_2}L(t)^\top\big\rangle \\
&\leq 2\big\langle \nabla\ell\big(A(0)\big), \nabla_{W_1}L(t)\nabla_{W_2}L(t)^\top\big\rangle + K\|\nabla_{W_2}L(t)W_2(t)^\top + W_1(t)\nabla_{W_1}L(t)^\top\|_F^2 \\
&\quad + 2\|\nabla\ell\big(A(\tau)\big) - \nabla\ell\big(A(0)\|_F \cdot \|\nabla_{W_1}L(t)\nabla_{W_2}L(t)^\top\|_F \\
&\quad + 4\tau^2\eta_t^2 K\|\nabla_{W_1}L(t)\nabla_{W_2}L(t)^\top\|_F^2 \\
&\quad + 4\tau\eta_t K\|\nabla_{W_2}L(t)W_2(t)^\top + W_1(t)\nabla_{W_1}L(t)^\top\|_F \cdot \|\nabla_{W_1}L(t)\nabla_{W_2}L(t)^\top\|_F \\
&\leq 2\big\langle \nabla\ell\big(A(0)\big), \nabla_{W_1}L(t)\nabla_{W_2}L(t)^\top\big\rangle + K\|\nabla_{W_2}L(t)W_2(t)^\top + W_1(t)\nabla_{W_1}L(t)^\top\|_F^2 \\
&\quad + 2\|\nabla\ell\big(A(\tau)\big) - \nabla\ell\big(A(0)\|_F \cdot \|\nabla_{W_1}L(t)\nabla_{W_2}L(t)^\top\|_F \\
&\quad + 4\eta_t^2 K\|\nabla_{W_1}L(t)\nabla_{W_2}L(t)^\top\|_F^2 \\
&\quad + 4\eta_t K\|\nabla_{W_2}L(t)W_2(t)^\top + W_1(t)\nabla_{W_1}L(t)^\top\|_F \cdot \|\nabla_{W_1}L(t)\nabla_{W_2}L(t)^\top\|_F,
\end{aligned}
\tag{84}
$$

where the last line is derived based on the fact that $\tau \in (0, 1]$.

Notice in equation 78, we have shown

$$
\begin{aligned}
2\big\langle \nabla\ell\big(A(0)\big), \nabla_{W_1}L(t)\nabla_{W_2}L(t)^\top\big\rangle &+ K\|\nabla_{W_2}L(t)W_2(t)^\top + W_1(t)\nabla_{W_1}L(t)^\top\|_F^2 \\
&\leq [\|\nabla_{W_1}L(t)\|_F^2 + \|\nabla_{W_2}L(t)\|_F^2] \cdot \big(\sqrt{2KL(t)} + \sigma_{\max}^2(\mathcal{T}_t)\big).
\end{aligned}
\tag{85}
$$

Moreover, in Lemma D.3, Lemma D.4 and Lemma D.5, we have shown

$$
\begin{aligned}
2\|\nabla_{W_1}L(t)\nabla_{W_2}L(t)^\top\|_F &\leq \|\nabla_{W_1}L(t)\|_F^2 + \|\nabla_{W_2}L(t)\|_F^2 \\
\|\nabla_{W_1}L(t)\nabla_{W_2}L(t)^\top\|_F &\leq 2K\sigma_{\max}(W(t))L(t) \\
\|\nabla_{W_2}L(t)W_2(t)^\top + W_1(t)\nabla_{W_1}L(t)^\top\|_F &\leq \sigma_{\max}^2(\mathcal{T}_t)\sqrt{2KL(t)} \\
\|\nabla\ell\big(A(s)\big) - \nabla\ell\big(A(0)\|_F &\leq \eta_t K\sqrt{2KL(t)}\sigma_{\max}^2(\mathcal{T}_t) + 2\eta_t^2 K^2\sigma_{\max}(W(t))L(t).
\end{aligned}
\tag{86}
$$
$$
\tag{87}
$$

Thus, one can further upper bound equation 84 as follows

$$
\begin{aligned}
&\left.\frac{d^2}{ds^2}M(s+\tau)\right|_{s=0} \\
&\leq 2\big\langle \nabla\ell\big(A(0)\big), \nabla_{W_1}L(t)\nabla_{W_2}L(t)^\top\big\rangle + K\|\nabla_{W_2}L(t)W_2(t)^\top + W_1(t)\nabla_{W_1}L(t)^\top\|_F^2 \\
&\quad + 2\|\nabla\ell\big(A(\tau)\big) - \nabla\ell\big(A(0)\big)\|_F \cdot \|\nabla_{W_1}L(t)\nabla_{W_2}L(t)^\top\|_F \\
&\quad + 4\eta_t^2 K\|\nabla_{W_1}L(t)\nabla_{W_2}L(t)^\top\|_F^2 \\
&\quad + 4\eta_t K\|\nabla_{W_2}L(t)W_2(t)^\top + W_1(t)\nabla_{W_1}L(t)^\top\|_F \cdot \|\nabla_{W_1}L(t)\nabla_{W_2}L(t)^\top\|_F \\
&\leq [\|\nabla_{W_1}L(t)\|_F^2 + \|\nabla_{W_2}L(t)\|_F^2] \cdot \big(\sqrt{2KL(t)} + \sigma_{\max}^2(\mathcal{T}_t)\big) \\
&\quad + \Big(\eta_t K\sqrt{2KL(t)}\sigma_{\max}^2(\mathcal{T}_t) + 2\eta_t^2 K^2\sigma_{\max}(W(t))L(t)\Big) \cdot [\|\nabla_{W_1}L(t)\|_F^2 + \|\nabla_{W_2}L(t)\|_F^2] \\
&\quad + 4\eta_t^2 K^2\sigma_{\max}(W(t))L(t) \cdot [\|\nabla_{W_1}L(t)\|_F^2 + \|\nabla_{W_2}L(t)\|_F^2] \\
&\quad + 2\eta_t K\sigma_{\max}^2(\mathcal{T}_t)\sqrt{2KL(t)} \cdot [\|\nabla_{W_1}L(t)\|_F^2 + \|\nabla_{W_2}L(t)\|_F^2].
\end{aligned} \tag{88}
$$

As a result, we can show

$$
\begin{aligned}
\langle g_t, H(\tau)g_t\rangle &= \frac{1}{\|\nabla L(t)\|_F^2}\cdot \left.\frac{d^2}{ds^2}M(s+\tau)\right|_{s=0} \\
&\leq K\sigma_{\max}^2(\mathcal{T}_t) + \sqrt{2KL(t)} + 6K^2\sigma_{\max}(W(t))L(t)\eta_t^2 + 3K\sigma_{\max}^2(\mathcal{T}_t)\sqrt{2KL(t)}\eta_t.
\end{aligned}
$$

$\square$

# E    PROOF OF THEOREM 3.2

In this section, we first introduce the generalized form of Theorem 3.2. Then, we provide a detailed proof.

**Theorem E.1** (Linear convergence of GD for Problem 2). *Assume the GD algorithm equation 8 is initialized such that $\alpha_1 > 0$. Pick any $0 < c < 1, d > 1$. Let $\eta_0^{(1)}$ be the unique positive solution of the following equation*

$$
\eta_0\big(\sqrt{2KL(0)} + 6K^2\beta_2 L(0)\eta_0^2 + K\exp(\sqrt{\eta_0})\alpha_2\big[1 + 3\sqrt{2KL(0)}\eta_0\big]\big) = 1, \tag{89}
$$

*and $\eta_0^{(2)}$ be the smallest positive solution of the following equation[3]*

$$
4KL(0)\eta_0^2 = (1 - \exp(-\eta_0^c)) \times (1 - \Delta). \tag{90}
$$

*Then, we define $\eta_{\max} = \min(\eta_0^{(1)}, \eta_0^{(2)}, \log\big(1 + \frac{\alpha_1}{2\alpha_2}\big)^{\frac{1}{c}})$. For any $\eta_0$ and $\eta_t$ such that $0 < \eta_0 \leq \eta_{\max}$ and $\eta_t$ satisfies*

$$
\eta_0 \leq \eta_t \leq \min\big((1 + \eta_0^d)^{\frac{t}{2}}\eta_0, \frac{1}{K_t}\big), \tag{91}
$$

*one can derive the following linear convergence rate for GD*

$$
L(t+1) \leq L(t)\bar{\rho}(\eta_t, t) \leq L(t)\bar{\rho}(\eta_0, 0) \leq L(0)\bar{\rho}(\eta_0, 0)^{t+1}, \tag{92}
$$

*where*

$$
\bar{\rho}(\eta_t, t) = 1 - 2\bar{\mu}\eta_t + \bar{\mu}\bar{K}_t\eta_t^2, \quad \bar{\mu} = \mu\big[\alpha_1 + 2\alpha_2\big(1 - \exp(\eta_0^c)\big)\big], \quad \Delta = (1 + \eta_0^d)\bar{\rho}(\eta_0, 0),
$$
$$
\bar{K}_t = \sqrt{2KL(0)\bar{\rho}(\eta_0, 0)^t} + 6K^2\beta_2 L(0)\eta_0^2\Delta^t + K\exp(\sqrt{\eta_0})\alpha_2\big[1 + 3\sqrt{2KL(0)\Delta^t}\eta_0\big].
$$

Notice Theorem 3.2 in §3.2 can be viewed as a special case of Theorem E.1 with $c = \frac{1}{2}, d = 2$.

---

[3]In the case when equation 90 does not have positive solution, we set $\eta_0^{(2)} = \infty$.

Before presenting the proof Theorem E.1, we first show that the constraints on $\eta_0$ do not induce an empty set, or equivalently $\eta_{\max} > 0$. Alongside, we provide several inequalities that are implied by the constraints on $\eta_0$, which the proof Theorem E.1 is relied on.

**Existence of $\eta_0$ .** To show the existence of $\eta_0$, it is equivalent to show that $\eta_{\max} > 0$. First, since $\alpha_1, \alpha_2$ are positive, we have $\log\left(1 + \frac{\alpha_1}{2\alpha_2}\right)^{\frac{1}{c}} > 0$. Moreover, one can see when $\eta_0 < \log\left(1 + \frac{\alpha_1}{2\alpha_2}\right)^{\frac{1}{c}}$, we have $\bar{\mu} > 0$.

Second, we show $\eta_0^{(1)} > 0$. The LHS of equation 89 increases as $\eta_0$ increases, and it equals zero as $\eta_0 = 0$. Thus, there exists a unique positive solution of equation 89, which is equivalent to $\eta_0^{(1)} > 0$. Notice

$$\bar{K}_0 = \sqrt{2KL(0)} + 6K^2\beta_2 L(0)\eta_0^2 + K\exp(\sqrt{\eta_0})\alpha_2\left[1 + 3\sqrt{2KL(0)}\eta_0\right]. \tag{93}$$

Therefore, $0 < \eta_0 \leq \eta_0^{(1)}$ implies $0 < \eta_0\bar{K}_0 \leq 1$ which is equivalent to $\eta_0 \leq \frac{1}{\bar{K}_0}$. This constraint further leads to $0 < \bar{\rho}(\eta_0, 0) < 1$ and $\Delta > 0$.

Finally, we show $\eta_0^{(2)} > 0$. Notice when $\eta_0 = 0$, the RHS and LHS of equation 90 both equal zero. Moreover, when $\eta_0 > 0$, one can rewrite equation 90 as follows

$$4KL(0)\eta_0^2 = (1 - \exp(-\eta_0^c)) \times (1 - \Delta)$$
$$\iff 4KL(0)\eta_0^2 = (1 - \exp(-\eta_0^c)) \times (2\bar{\mu}\eta_0 - \bar{\mu}\bar{K}_0\eta_0^2 - \eta_0^d\bar{\rho}(\eta_0, 0))$$
$$\iff 4KL(0)\eta_0^{1-c} = \frac{1 - \exp(-\eta_0^c)}{\exp(-\eta_0^c)} \times (2\bar{\mu} - \bar{\mu}\bar{K}_0\eta_0 - \eta_0^{d-1}\bar{\rho}(\eta_0, 0)). \tag{94}$$

Then, we study the order of both sides of equation 94 in terms of $\eta_0$ in the regime where $0 < \eta_0 \leq \min\left(\log\left(1 + \frac{\alpha_1}{2\alpha_2}\right)^{\frac{1}{c}}, \frac{1}{\bar{K}_0}\right)$. Since $0 < c < 1$, and the LHS of equation 94 is of order $\Theta(\eta_0^{1-c})$, it decreases monotonically to zero as $\eta_0$ approaches zero. The RHS of equation 94 is the product of two terms, i.e., $\frac{1-\exp(-\eta_0^c)}{\exp(-\eta_0^c)}$ and $2\bar{\mu} - \bar{\mu}\bar{K}_0\eta_0 - \eta_0^{d-1}\bar{\rho}(\eta_0, 0)$. We notice $\eta_0^c$ approaches zero as $\eta_0$ decreases to zero. Thus, $\frac{1-\exp(-\eta_0^c)}{\exp(-\eta_0^c)}$ converges to one as $\eta_0$ decreases to zero. Moreover, when $\eta_0 \leq \min\left(\log\left(1 + \frac{\alpha_1}{2\alpha_2}\right)^{\frac{1}{c}}, \frac{1}{\bar{K}_0}\right)$, we have $\bar{\mu} > 0$ and $0 < \bar{\rho}(\eta_0, 0) < 1$. Therefore, the RHS of equation 94 is of order $\Theta(1)$. As a result, when $\eta_0 > 0$ is sufficiently small, one has

$$4KL(0)\eta_0^2 < (1 - \exp(-\eta_0^c)) \times (1 - \Delta). \tag{95}$$

Moreover, if equation 90 has positive roots, and we use $\eta_0^{(2)}$ to denote its smallest positive root. The following holds for all $0 < \eta_0 \leq \eta_0^{(2)}$

$$4KL(0)\eta_0^2 \leq (1 - \exp(-\eta_0^c)) \times (1 - \Delta) \tag{96}$$

If equation 90 does not have positive root, then equation 95 holds for all positive $\eta_0$.

To summarize, we have shown that $\eta_{\max} > 0$, and the $\eta_0$ always exists. Moreover, when $\eta_0$ satisfies $0 < \eta_0 < \eta_{\max}$, the following holds

$$\bar{\mu} > 0, \quad 0 < \bar{\rho}(\eta_0, 0), \Delta < 1,$$
$$4KL(0)\eta_0^2 \leq (1 - \exp(-\eta_0^c)) \times (1 - \Delta). \tag{97}$$

Now we present the proof of Theorem E.1.

*Proof.* We employ an induction-based approach to prove Theorem E.1 by iteratively showing the following properties hold for all iteration $t$ when $\eta_0$ and $\eta_t$ satisfy the constraints in Theorem E.1.

- $A_1(t) : L(t) \leq L(t-1)\bar{\rho}(\eta_{t-1}, t-1) \leq L(t-1)\bar{\rho}(\eta_0, 0)$.

- $A_2(t) : \beta_1 \leq \sigma_{\min}(W(t)) \leq \sigma_{\max}(W(t)) \leq \beta_2$.

- $A_3(t) : \|D(t) - D(0)\|_F \le \frac{2K\eta_0^2 \alpha_2(0) \exp(\eta_0^c) L(0)}{1-\Delta}$.

- $A_4(t) : \alpha_1 + 2\alpha_2\big(1 - \exp(\eta_0^c)\big) \le \sigma_{\min}^2(\mathcal{T}_t) \le \sigma_{\max}^2(\mathcal{T}_t) \le \alpha_2 \exp(\eta_0^c)$.

Assume $A_1(k), A_2(k), A_3(k), A_4(k)$ hold at iteration $k = 1, 2, \cdots, t$, then we show they all hold for iteration $t+1$.

**Prove $A_1(t+1)$ hold.**

We first show that under the constraints in Theorem E.1 and the induction assumption, one can lower bound and upper bound $\mu_t$ and $K_t$ using $\bar{\mu}$ and $\bar{K}_t$ respectively, which is characterized by the following lemma.

**Lemma E.1.** *The following lower bound and upper bound on $\mu_t$ and $K_t$ hold respectively*

$$\bar{\mu} \le \mu_t, \quad K_t \le \bar{K}_t. \tag{98}$$

The proof of the above lemma can be found at the end of Appendix E.

In Theorem 3.1, we have shown that the local PL inequality and Descent lemma hold with local PL constant $\mu_t$ and local smoothness constant $K_t$

$$L(t+1) \le L(t) - \big(\eta_t - \frac{K_t \eta_t^2}{2}\big)\|\nabla L(t)\|_F^2, \quad \frac{1}{2}\|\nabla L(t)\|_F^2 \ge \mu_t L(t). \tag{99}$$

Therefore, one has

$$
\begin{aligned}
L(t+1) &\le L(t) - \big(\eta_t - \frac{K_t \eta_t^2}{2}\big)\|\nabla L(t)\|_F^2 \\
&\le L(t) - 2\mu_t\big(\eta_t - \frac{K_t \eta_t^2}{2}\big)L(t) && \text{Under the constraints } 0 < \eta_t < \frac{1}{K_t} \\
&\le L(t) - 2\bar{\mu}\big(\eta_t - \frac{K_t \eta_t^2}{2}\big)L(t) && \text{Lemma E.1} \\
&= (1 - 2\bar{\mu}\eta_t + \bar{\mu}K_t\eta_t^2)L(t) \\
&\le (1 - 2\bar{\mu}\eta_t + \bar{\mu}\bar{K}_t\eta_t^2)L(t) := \bar{\rho}(\eta_t, t)L(t) && \text{Lemma E.1}. \tag{100}
\end{aligned}
$$

Finally, we show $\rho(\eta_t, t) \le \rho(\eta_0, 0)$.

$$
\begin{aligned}
\rho(\eta_t, t) &= 1 - 2\bar{\mu}\eta_t + \bar{\mu}\bar{K}_t\eta_t^2 \\
&\le 1 - 2\bar{\mu}\eta_0 + \bar{\mu}\bar{K}_t\eta_0^2 && \text{Use } \eta_0 \le \eta_t \le \frac{1}{K_t} \\
&\le 1 - 2\bar{\mu}\eta_0 + \bar{\mu}\bar{K}_0\eta_0^2 := \rho(\eta_0, 0) && \text{Use } \bar{K}_t \le \bar{K}_0. \tag{101}
\end{aligned}
$$

Therefore, $A_1(t+1)$ holds.

**Prove $A_2(t+1)$ hold.**

Since we have shown $A_1(t+1)$ holds, one has $L(t+1) \le L(0)$. Moreover, based on the assumption that $\ell(W)$ is $\mu$-strongly convex and $K$-smooth, one has the following inequality

$$\frac{\mu}{2}\|W(t+1) - W^*\|_F^2 \le \ell(t+1) = L(t+1) \le \frac{K}{2}\|W(t+1) - W^*\|_F^2. \tag{102}$$

Then we can show $\sigma_{\max}(W(t+1)) \le \beta_2$ as follows

$$
\begin{aligned}
\sigma_{\max}(W(t+1)) &= \sigma_{\max}(W(t+1) - W^* + W^*) \\
&\le \sigma_{\max}(W^*) + \|W(t+1) - W^*\|_2 && \text{Weyl's inequality} \\
&\le \sigma_{\max}(W^*) + \|W(t+1) - W^*\|_F \\
&\le \sigma_{\max}(W^*) + \sqrt{\frac{2}{\mu}L(t+1)} \\
&\le \sigma_{\max}(W^*) + \sqrt{\frac{2}{\mu}L(0)}. && \text{Use } L(t+1) \le L(0) \tag{103}
\end{aligned}
$$

For $\beta_1 \leq \sigma_{\min}(W(t+1))$, same result has been derived in Min et al. (2023). We refer the readers to Appendix B in Min et al. (2023) for details.

**Prove $A_3(t+1)$ hold.**

We first present the following lemma that bounds $\|D(k+1) - D(k)\|_F$ for all $k$.

**Lemma E.2.** *One has the following upper bound on $\|D(k+1) - D(k)\|_F$*

$$\|D(k+1) - D(k)\|_F \leq 2K\eta_k^2 \sigma_{\max}^2(\mathcal{T}_k) L(k) . \tag{104}$$

The proof of the above lemma can be found at the end of this section.

Based on Lemma E.2, one can show that $A_3(t+1)$ holds

$$
\begin{aligned}
\|D(t+1) - D(0)\|_F &\leq \sum_{k=0}^{t} \|D(k+1) - D(k)\|_F \\
&\leq \sum_{k=0}^{t} 2K\eta_k^2 \sigma_{\max}^2(\mathcal{T}_k) L(k) && \text{Lemma E.2} \\
&\leq \sum_{k=0}^{t} 2K\eta_k^2 \sigma_{\max}^2(\mathcal{T}_k) L(0) \bar{\rho}(\eta_0, 0)^k && \text{Use } A_1(k), \forall k = 1, \cdots, t \\
&\leq \sum_{k=0}^{t} 2K\eta_k^2 \alpha_2 \exp(\eta_0^c) L(0) \bar{\rho}(\eta_0, 0)^k && \text{Use } A_4(k), \forall k = 1, \cdots, t \\
&\leq \sum_{k=0}^{t} 2K(1+\eta_0^d)^k \eta_0^2 \alpha_2 \exp(\eta_0^c) L(0) \bar{\rho}(\eta_0, 0)^k && \text{Use } \eta_k \leq (1+\eta_0^d)^{\frac{k}{2}} \eta_0 \\
&= 2KL(0) \exp(\eta_0^c) \eta_0^2 \alpha_2 \sum_{k=0}^{t} \Delta^k && \Delta = (1+\eta_0^d) \bar{\rho}(\eta_0, 0) \\
&\leq \frac{2K\eta_0^2 \alpha_2 \exp(\eta_0^c) L(0)}{1 - \Delta} . && 0 < \Delta < 1 \tag{105}
\end{aligned}
$$

**Prove $A_4(t+1)$ hold.**

We first present the following two lemmas which will be used to prove that $A_4(t+1)$ hold.

**Lemma E.3.** *One can use $\alpha_1, \alpha_2$ to lower and upper bound the singular values of $\mathcal{T}_0$*

$$\alpha_1 \leq \sigma_{\min}^2(\mathcal{T}_0) \leq \sigma_{\max}^2(\mathcal{T}_0) \leq \alpha_2 . \tag{106}$$

**Lemma E.4.** *One can bound the deviation of the singular values of $\mathcal{T}_k$ using the deviation of the imbalance $\|D(k) - D(0)\|_F$*

$$\sigma_{\min}^2(\mathcal{T}_k) \geq \alpha_1 - 4\|D(k) - D(0)\|_F := \mathcal{T}_k^L . \tag{107}$$
$$\sigma_{\max}^2(\mathcal{T}_k) \leq \alpha_2 + 2\|D(k) - D(0)\|_F := \mathcal{T}_k^U . \tag{108}$$

The proof of Lemma E.3 and Lemma E.4 can be in Xu et al. (2023), Appendix C.

Notice $\mathcal{T}_{t+1}^L + 2\mathcal{T}_{t+1}^U = \alpha_1 + 2\alpha_2$. Therefore, if one can show

$$\mathcal{T}_{t+1}^U \leq \exp(\eta_0^c) \alpha_2 . \tag{109}$$

Then, the following holds directly

$$\sigma_{\max}^2(\mathcal{T}_{t+1}) \leq \mathcal{T}_{t+1}^U \leq \exp(\eta_0^c) \alpha_2 , \tag{110}$$
$$\sigma_{\min}^2(\mathcal{T}_{t+1}) \geq \mathcal{T}_{t+1}^L = \alpha_1 + 2\alpha_2 - 2\mathcal{T}_{t+1}^U \geq \alpha_1 + 2\alpha_2 \left(1 - \exp(\eta_0^c)\right) . \tag{111}$$

Therefore, it suffices to show equation 109 holds. We start from Lemma E.4

$$
\begin{aligned}
\mathcal{T}_k^{\mathrm{U}} &= \alpha_2 + 2\|D(k) - D(0)\|_F \\
&\leq \alpha_2 + \frac{4KL(0)\eta_0^2\alpha_2(0)\exp(\eta_0^c)}{1-\Delta} && \text{Use } A_3(t+1) \\
&\leq \alpha_2 + (1 - \exp(-\eta_0^c)) \times (1-\Delta) \cdot \frac{\alpha_2\exp(\eta_0^c)}{1-\Delta} && \text{Equation 95} \\
&= \exp(\eta_0^c)\alpha_2\,.
\end{aligned}
\tag{112}
$$

$\square$

Now, we present the proof of lemmas used in the proof of Theorem E.1. All lemmas presented below are based on the assumption that $A_1(k), A_2(k), A_3(k), A_4(k)$ hold for all iterations $k = 1, 2, \cdots, t$ and the constraints presented in Theorem E.1. For convenience, we do not state these assumptions and constraints repetitively.

**Lemma E.1.** *The following lower bound and upper bound on $\mu_t$ and $K_t$ hold respectively*

$$
\bar{\mu} \leq \mu_t\,, \quad K_t \leq \bar{K}_t\,.
\tag{113}
$$

*Proof.* We start with the lower bound on $\mu_t$. Due to the assumption that $A_4(t)$ hold, one has the following lower bound $\mu_t$

$$
\mu_t = \mu\sigma_{\min}^2(\mathcal{T}_t) \geq \mu\alpha_1\,.
\tag{114}
$$

For the upper bound on $K_t$, we first show that based on the assumption that $A_1(k)$ hold for all $k \leq t$, one has

$$
L(t) \leq L(t-1)\bar{\rho}(\eta_0, 0) \leq L(0)\bar{\rho}(\eta_0, 0)^t\,.
\tag{115}
$$

Then, based on equation 115, $A_4(t)$ and the constraint that $\eta_t \leq (1 + \eta_0^d)^{\frac{t}{2}}\eta_0$, we can derive the following upper bound on $K_t$

$$
\begin{aligned}
K_t &= K\sigma_{\max}^2(\mathcal{T}_t) + \sqrt{2KL(t)} + 6K^2\sigma_{\max}(W(t))L(t)\eta_t^2 + 3K\sigma_{\max}^2(\mathcal{T}_t)\sqrt{2KL(t)}\eta_t \\
&\leq K\alpha_2\exp(\eta_0^c) + \sqrt{2KL(0)\bar{\rho}(\eta_0, 0)^t} + 6K^2\beta_2 L(0)\bar{\rho}(\eta_0, 0)^t\eta_t^2 \\
&\quad + 3K\alpha_2\exp(\eta_0^c)\sqrt{2KL(0)\bar{\rho}(\eta_0, 0)^t}\eta_t \\
&\leq K\alpha_2\exp(\eta_0^c) + \sqrt{2KL(0)\bar{\rho}(\eta_0, 0)^t} + 6K^2\beta_2 L(0)\bar{\rho}(\eta_0, 0)^t(1 + \eta_0^d)^t\eta_0^2 \\
&\quad + 3K\alpha_2\exp(\eta_0^c)\sqrt{2KL(0)\bar{\rho}(\eta_0, 0)^t}(1 + \eta_0^d)^{\frac{t}{2}}\eta_0 && \text{Use } \eta_t \leq (1+\eta_0^d)^{\frac{t}{2}}\eta_0 \\
&= \sqrt{2KL(0)\bar{\rho}(\eta_0, 0)^t} + 6K^2\beta_2 L(0)\eta_0^2\Delta^t + K\exp(\sqrt{\eta_0})\alpha_2\big[1 + 3\sqrt{2KL(0)\Delta^t}\eta_0\big]\,,
\end{aligned}
\tag{116}
$$

where the last line follows the definition of $\Delta = (1 + \eta_0^d)\bar{\rho}(\eta_0, 0)$.  $\square$

**Lemma E.2.** *One has the following upper bound on $\|D(k+1) - D(k)\|_F$*

$$
\|D(k+1) - D(k)\|_F \leq 2K\eta_k^2\sigma_{\max}^2(\mathcal{T}_k)L(k)\,.
\tag{117}
$$

*Proof.* In equation 8 and equation 9, we have

$$
W_1(k+1) = W_1(k) - \eta_k\nabla\ell(k)W_2(k)\,, \quad W_2(k+1) = W_2(k) - \eta_k\nabla\ell(k)^\top W_1(k)\,.
\tag{118}
$$

There, we can compute $D(k+1) - D(k)$ as follows

$$
\begin{aligned}
D(k+1) - D(k) &= W_1(k+1)^\top W_1(k+1) - W_2(k+1)^\top W_2(k+1) \\
&\quad - W_1(k)^\top W_1(k) + W_2(k)^\top W_2(k) \\
&= \big(W_1(k) - \eta_k\nabla\ell(k)W_2(k)\big)^\top\big(W_1(k) - \eta_k\nabla\ell(k)W_2(k)\big) \\
&\quad - \big(W_2(k) - \eta_k\nabla\ell(k)^\top W_1(k)\big)^\top\big(W_2(k) - \eta_k\nabla\ell(k)^\top W_1(k)\big) \\
&\quad - W_1(k)^\top W_1(k) + W_2(k)^\top W_2(k) \\
&= \eta_k^2\big(W_2(k)^\top\nabla\ell(k)^\top\nabla\ell(k)W_2(k) - W_1(k)^\top\nabla\ell(k)^\top\nabla\ell(k)W_1(k)\big)\,.
\end{aligned}
\tag{119}
$$

Based on the above equation, one can bound $\|D(k+1)-D(k)\|_F$ as follows

$$
\begin{aligned}
\|D(k+1)-D(k)\|_F &= \eta_k^2 \|W_2(k)^\top \nabla\ell(k)^\top \nabla\ell(k) W_2(k) - W_1(k)^\top \nabla\ell(k)^\top \nabla\ell(k) W_1(k)\|_F \\
\text{Property of norm } &\leq \eta_k^2 \|W_2(k)^\top \nabla\ell(k)^\top \nabla\ell(k) W_2(k)\|_F + \eta_k^2 \|W_1(k)^\top \nabla\ell(k)^\top \nabla\ell(k) W_1(k)\|_F \\
\text{equation } 30 \ &\leq \eta_k^2 \sigma_{\max}^2(W_2(k))\|\nabla\ell(k)\|_F^2 + \eta_k^2 \sigma_{\max}^2(W_1(k))\|\nabla\ell(k)\|_F^2 \\
&= \eta_k^2 \sigma_{\max}^2(\mathcal{T}_k)\|\nabla\ell(k)\|_F^2 \\
K\text{-smooth of } \ell \ &\leq 2K\eta_k^2 \sigma_{\max}^2(\mathcal{T}_k)L(k) \,.
\end{aligned}
\tag{120}
$$

$\square$

# F  VERIFICATION OF THE ASSUMPTION $\alpha_1 > 0$

In this section, we provide two conditions that ensure $\alpha_1 > 0$.

In Min et al. (2021), the authors show the following lemma which guarantees $\alpha_1 > 0$.

**Lemma F.1** (Lemma 1 in (Min et al., 2021)). *Let $W_1(0), W_2(0)$ are initialized entry-wise i.i.d. from $\mathcal{N}(0, \frac{1}{h^{2p}})$ with $\frac{1}{4} \leq p \leq \frac{1}{2}$. For $\forall \delta > 0$ and $h \geq poly(n, m, \frac{1}{\delta})$, with probability $1 - \delta$ over random initialization with $W_1(0), W_2(0)$, the following holds*

$$
\alpha_1 \geq h^{1-2p} \,.
\tag{121}
$$

The above theorem states when Problem 2 is **sufficiently overparametrized**, i.e., $h \geq \text{poly}(n, m, \frac{1}{\delta})$, Gaussian initialization with proper variance ensures $\alpha_1$ has a positive lower bound $h^{1-2p}$. Moreover, the lower bound increases as $h$ increases.

Next, we are going to show with **mild** overparametrization, one can ensure $\alpha_1 > 0$.

**Lemma F.2** (Mild overparametrization ensures $\alpha_1 > 0$). *Let $W_1(0), W_2(0)$ are initialized entry-wise i.i.d. from a continuous distribution $\mathbb{P}$. When $h \geq m + n$, the following holds almost surely over random initialization with $W_1(0), W_2(0)$*

$$
\alpha_1 > 0 \,.
\tag{122}
$$

Compared with Lemma F.1, Lemma F.2 considers a wider range of distributions that include Gaussian distribution and uniform distribution. Thus, commonly used random initialization schemes, such as Xavier initialization (Glorot & Bengio, 2010) and He initialization (He et al., 2015), lead to $\alpha_1 > 0$. Moreover, the requirement of overparametrization in Lemma F.2 is mild compared with the one in Lemma F.1, i.e., $h \geq m + n$ versus $h \geq \text{poly}(n, m, \frac{1}{\delta})$. As a result, Lemma F.2 can be applied to more general overparametrization. On the other hand, the conclusion of Lemma F.2 is weaker than Lemma F.1 in the sense that Lemma F.2 only proves $\alpha_1 > 0$ but do not characterize its magnitude.

Before presenting the proof of Lemma F.2, we first present two lemmas that will be used in the proof.

**Lemma F.3.** *Let $A \in \mathbb{R}^{h \times n}, h \geq n$ be a random matrix with entry-wise drawn i.i.d. from a continuous distribution $\mathbb{P}$. Then $A$ is of full column rank almost surely.*

We refer the readers to (Vershynin, 2018) for detailed proof.

**Lemma F.4.** *A sufficient condition for $\alpha_1 > 0$ is $\sigma_{m+n}(D(0)) > 0$.*

The proof of this lemma can be found in (Min et al., 2021).

Now we present the proof of Lemma F.2

*Proof.* Based on Lemma F.4, it suffices to show that one almost surely has $\sigma_{m+n}(D(0)) > 0$ over random initialization with $W_1(0), W_2(0)$. We use proof by contradiction. Assume $\sigma_{m+n}(D(0)) = 0$, then one has $\dim(\ker D(0)) \geq h - n - m + 1$.

On the other hand, Lemma F.3 implies with probability one, $[W_1^\top(0), W_2^\top(0)] \in \mathbb{R}^{h \times (n+m)}$ is of full column rank. Our next step is to show $\dim(\ker D(0)) \leq h - n - m$. If this is true, then there is a contradiction. Thus, one directly has $\sigma_{m+n}(D(0)) > 0$.

For any $v \in \mathbb{R}^h$ that satisfies $D(0)v = 0$, we can write this equation as follows

$$D(0)v = 0 \Leftrightarrow [W_1^\top(0), W_2^\top(0)] \begin{bmatrix} W_1(0) \\ -W_2(0) \end{bmatrix} v = 0 \tag{123}$$

Since $[W_1^\top(0), W_2^\top(0)]$ is of full column rank, the above equation is equivalent to

$$\begin{bmatrix} W_1(0) \\ -W_2(0) \end{bmatrix} v = 0 \,, \tag{124}$$

and $\dim(\ker D(0)) \leq h - n - m$. $\qquad\qquad\square$

## G  SIMULATION

In this section, we first present empirical evidence that Theorem 3.2 provides a good characterization of the actual convergence rate under different initializations. Moreover, we compare the convergence rate of GD using the adaptive step size proposed in equation 19, in Section 3.3 of Xu et al. (2023), and backtracking line search. Throughout the simulations, we train two-layer linear networks on the square loss.

$$\min_{W_1, W_2} \frac{1}{2} \|Y - X W_1 W_2^\top\|_F^2 \,, \tag{125}$$

where $X, Y \in \mathbb{R}^{10 \times 10}$ are data matrices and $W_1, W_2 \in \mathbb{R}^{10 \times h}$ are the weights. This can be viewed as a two-layer linear network with input and output dimensions 20 and the width of the hidden layer to be $h$. Throughout the simulations, we choose $h \in \{500, 1000, 4000\}$. We choose $c = 0.5, d = 1.01$ in Theorem E.1. The initialization of the weights and generation of data matrices are as follows: $W_1(0), W_2(0) \in \mathbb{R}^{10 \times h}$, and have entry-wise i.i.d. samples drawn from $\mathcal{N}(0, 1)$. We generate $X$ as a random orthogonal matrix, and $Y = X W_1(0) W_2(0) + \sigma^2 \epsilon$ where $\epsilon \in \mathbb{R}^{10 \times 10}$ and are entry-wise i.i.d. samples drawn from $\mathcal{N}(0, 1)$. When $\sigma^2$ is large, the initial loss is large, thus the margin is small. Moreover, we experimentally observe that the initial imbalance grows w.r.t. $h$. The choices of $h$ and $\sigma$ allow us to test our results in different regimes.

### G.1  EVALUATION OF THE TIGHTNESS OF THE THEORETICAL BOUND ON THE CONVERGENCE RATE

Figure 1 compares the actual convergence rate of $L(t)$ versus the theoretical upper bound in §3.2 for different choices of $\sigma$ and $h$, and dissimilar $\frac{\alpha_0}{\beta_0}$. In all cases, the theoretical upper bound follows the actual loss well. Moreover, we observe for each adaptive step size scheme, the theoretical bounds and the actual rate of convergence become slower as $\frac{\alpha_0}{\beta_0}$ decreases. This is because our bounds on the local rate of convergence depend on $\frac{\alpha_0}{\beta_0}$, and the smaller $\frac{\alpha_0}{\beta_0}$, the slower the convergence rate. Finally, when $\sigma^2 = 1$, there is sufficient imbalance and the initial margin is zero which violates the assumptions in Arora et al. (2018); Du et al. (2018a) but GD still enjoys linear convergence. Thus, our theory applies beyond the regime of Arora et al. (2018); Du et al. (2018a).

### G.2  COMPARISON WITH PRIOR WORK AND BACKTRACKING LINE SEARCH

In this subsection, we compare the adaptive step sizes proposed in Xu et al. (2023), backtracking line search with the step sizes proposed in equation 19 with $h(\eta_t) = \rho(\eta_t, t)$. We set the hyperparameters of the adaptive step size scheme proposed in Xu et al. (2023) to be $c_1 = 0.5, c_2 = 1.5$, which is the same setting in their simulations. For backtracking line search, the algorithm is described as follows:

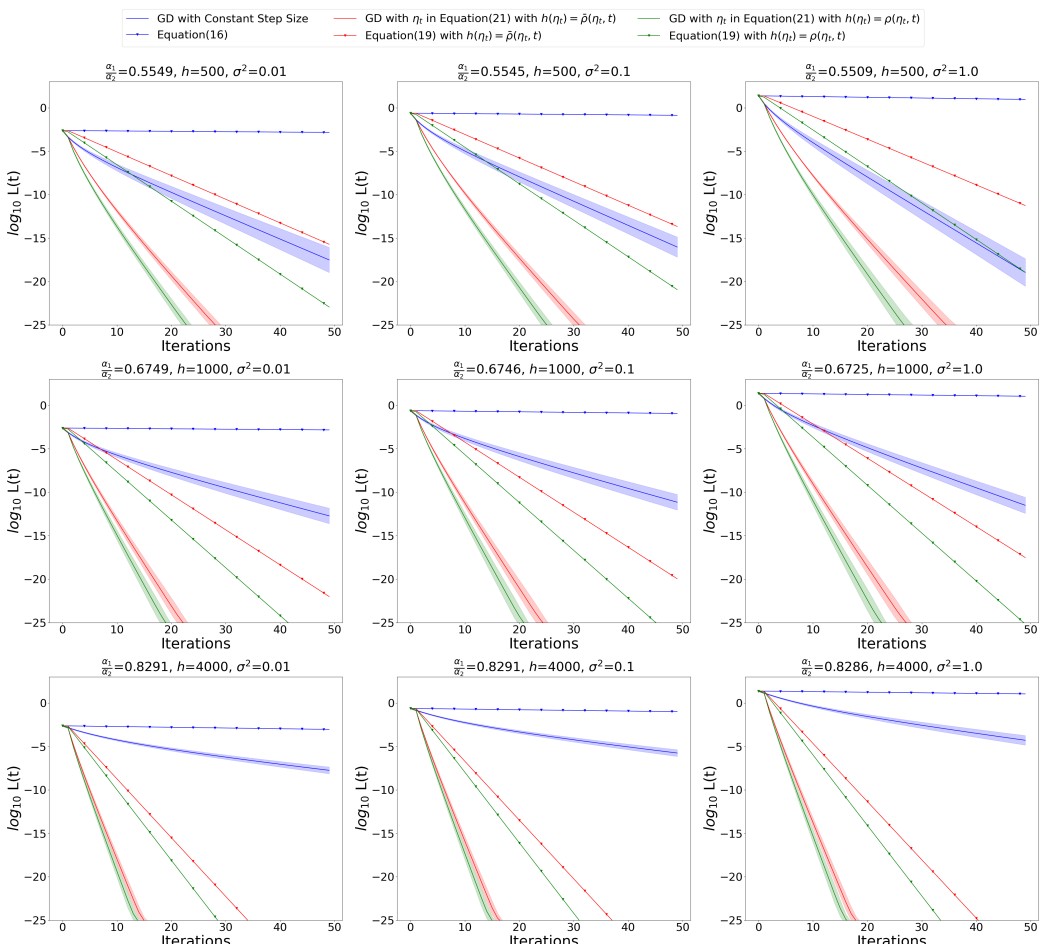

Figure 1: Tightness of the theoretical upper bound versus reconstruction error $L(t)$ for different choices of step size in §3.2, shown in different colors. We run the simulations for nine different settings of initialization and data generation. For each setting, we repeat the simulation thirty times. The triangle lines represent the theoretical upper bound on the training loss in equation 17 and equation 20. The solid lines represent the mean of the $\log_{10}$ of the reconstruction error $L(t)$. The shaded area is the mean of $\log_{10} L(t)$ plus and minus one standard deviation.

---

**Algorithm 1** Backtracking Line Search.

---

**Given** Data matrices $X, Y$, initialization $W_1(0), W_2(0)$, and hyperparameters $\eta_{bt}, \tau, \gamma$.
**Result** $W_1^*, W_2^*$ that minimize $L(W_1, W_2) = \frac{1}{2}\|Y - XW_1W_2^\top\|_F^2$.
**for** $t = 0, 1, \cdots T$ **do**
    $\eta_t = \eta_{bt}$
    **While** $L\big(W_1(t) - \eta_t \nabla_{W_1} L(t), W_2(t) - \eta_t \nabla_{W_2} L(t)\big) > L(t) - \gamma\|\nabla L(t)\|_F^2$
    $\eta_t = \tau\eta_t$
    $W_1(t+1) = W_1(t) - \eta_t \nabla_{W_1} L(t), W_2(t+1) = W_2(t) - \eta_t \nabla_{W_2} L(t)$.
**end for**

---

In the simulation, we choose $\tau = 0.1$ and $\gamma = 0.9$.

Figure 2 shows that the step size choice proposed in equation 19 achieves the fastest convergence compared with Xu et al. (2023) and backtracking line search in different settings. In all settings, the adaptive step size scheduler proposed in this work outperforms the other two methods. The reason is the following. For the adaptive step size scheduler in this work, the step size at each iteration

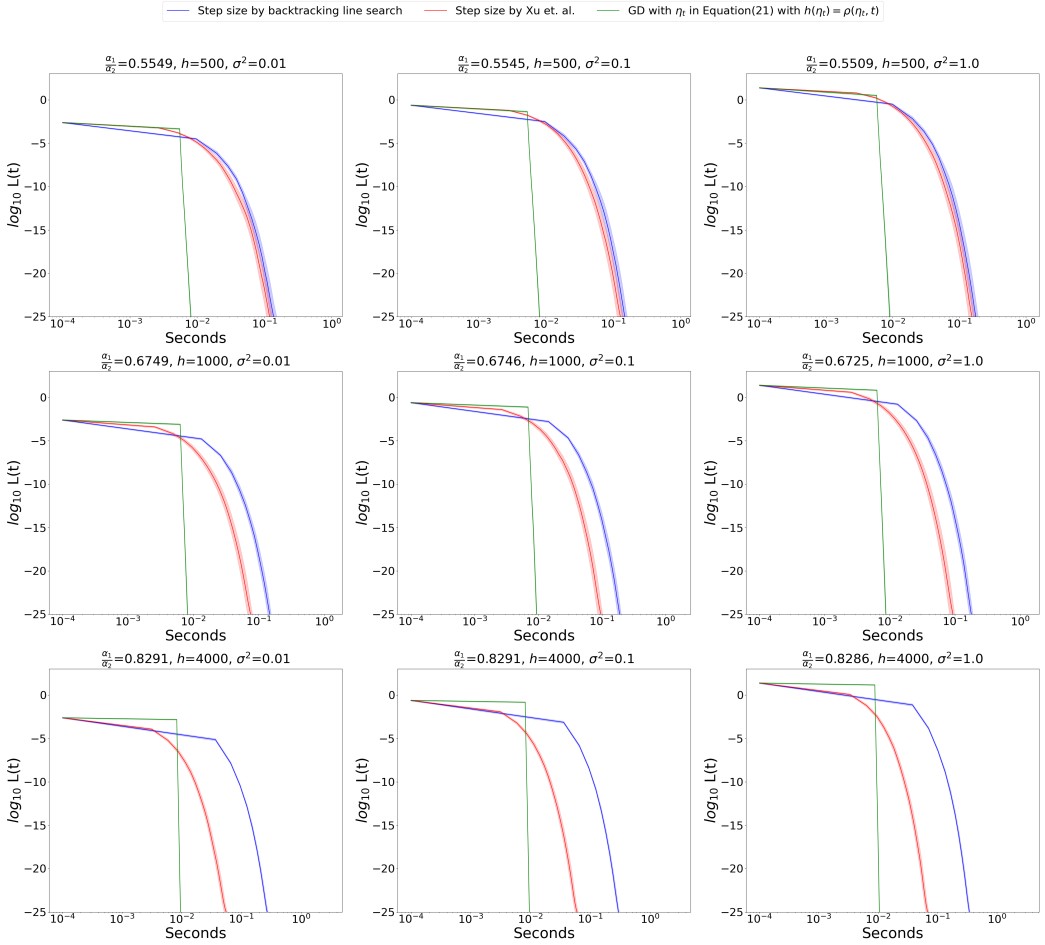

Figure 2: Evolution of the loss and of the step size for different choices of the step size schedule under different initialization and data generation. We run the simulations thirty times. For each setting, we repeat the simulation thirty times. The solid lines represent the mean of $\log_{10}$ of the reconstruction error $L(t)$. The shaded area is the mean of $\log_{10} L(t)$ plus and minus one standard deviation.

has closed form (See equation 19), thus the time for picking the optimal step size per iteration is negligible. The only time-consuming part is to find $\eta_0$ since one needs to solve equation 89 and equation 90 to get $\eta_{\max}$. For the step size proposed in Xu et al. (2023), the algorithm consists of solving a third-order polynomial at each iteration, which results in larger computational time. Moreover, the adaptive scheduler proposed in our work follows a sharper characterization of the local convergence rate than Xu et al. (2023), and the adaptive step size scheduler in this work theoretically converges of order $\frac{\alpha_1}{\alpha_2}$ faster than the one proposed in Xu et al. (2023). For the backtracking line search algorithm, since the algorithm iteratively searches for the step size at each iteration. Therefore, the time cost for each iteration is high as well.