# OpenReview forum: "A LOCAL POLYAK-ŁOJASIEWICZ AND DESCENT LEMMA OF GRADIENT DESCENT FOR OVERPARAMETERIZED LINEAR MODELS"
_ICLR.cc/2024/Conference — Submitted to ICLR 2024_

### Official Review · Reviewer_7VmP · 2023-10-26

**Soundness:** 4 excellent
**Presentation:** 3 good
**Contribution:** 3 good
**Rating:** 6
**Confidence:** 4

**Summary:**

Well-structured and well-presented. However, I suspect that the result can be incremental: there have been a lot of results applying the PL inequality to get convergence of neural networks, e.g. [Nguyen & Mondelli, 2020] (which is not cited). Also, they do not really discuss the requirement \alpha_1 > 0 much. It does not seem like a purely technical requirement. I suspect it might fail in reality, at least, sometimes.

The paper considers a general optimization problem for a two-layered linear network and aims to prove that GD converges to a minimum with a linear rate under some constraints on learning rate. Curiously, the learning rate can even increase throughout training.

The paper starts with a review of the classical linear convergence analysis for linear models by Polyak. This analysis stems on two ingredients: 1) Descent lemma, and 2) PL-inequality. However, neither PL-inequality, nor smoothness inequality which the Descent lemma is based on, cannot hold globally for a multi-layered linear model. The paper presents generalizations of both results with "local" smoothness constants. The local smoothness constants allow for bounds which are sufficient to derive linear convergence of GD under some (time-dependent) learning rate constraints.

The paper is very well-written. The first diagram is weird (and not really the way to do things... e.g. putting NTK in the 'finite step size' category is weird).

The paper contains no experimental validation for the main linear convergence result.

Questions: (1) Th.3.2 requires \alpha_1 > 0; what is the probability for this requirement to fail for the standard [Glorot & Bengio, 2010] initialization? (2) How could we estimate \eta_\max? How small could it be? What does it depend on?

**Strengths:**

Interesting theoretical result, useful, sound. Good discussion.

**Weaknesses:**

Literature review misses some references of closely-related (and even possibly overlapping works), e.g [Nguyen-Mondelli, 2020], [Radhakrishnan-Belkin-Uhler, 2021]. One really needs to clarify the novelty compared to existing results.

**Questions:**

(1) Th.3.2 requires \alpha_1 > 0; what is the probability for this requirement to fail for the standard [Glorot & Bengio, 2010] initialization?

(2) How could we estimate \eta_\max? How small could it be? What does it depend on?

(3) Novelty wrt existing literature.

---

> ### Author Response · Authors · 2023-11-23
>
> To [7VmP], we thank the reviewer for the thoughtful comments. We address the questions below.
> 1. Verification of $\alpha_1>0$.
> Please see our response to all reviewers.
> 2. Missing experimental validation for the linear convergence rate.
> Please see Appendix G.2 where we numerically verify that Problem(2) trained via gradient descent achieves linear convergence under different choices of the hyperparameters. Moreover, we show numerically the theoretical upper bound follows the actual loss well.
> 2. How to estimate \eta_\max?
> In Appendix E, we show $\eta_\max=\min(\eta_0^{(1)}, \eta_0^{(2)}, \log(1+\frac{\alpha_1}{2\alpha_2}))^{1/c}$. Thus, $\eta_\max$ depends on initial weights, $c, d, \mu, K$, and $L(0)$. One can see that if $L(0), K, \beta_2 $and $c, d$ increase or $\mu, \frac{\alpha_1}{\alpha_2}$ decrease, then $\eta_\max$ will decrease accordingly. To numerically solve for $\eta_\max$, one can use existing equation solver to solve eq(87) and eq(88) to get $\eta_0^{(1)}, \eta_0^{(2)}$. Then, $\eta_\max$ can be estimated by the minimum value by its definition.
> 3. Novelty compared with existing literature.
> We thank the reviewer for pointing us to these results (Nguyen et al., 2020; Radhakrishanan et al., 2020). We have added (Nguyen et al., 2020) in the "Related Work" section. In (Radhakrishanan et al., 2020), the authors study the convergence of autoencoder which is less relevant to our work. Thus, we won't cite it in the paper. We clarify the novelty of this work compared to existing results as follows. Existing results that study the convergence of deep linear networks (Arora et al., 2018; Du et al., 2018a) or deep nonlinear networks (Nguyen et al., 2020; Du et al., 2018; Jacot et al., 2018; Liu et al., 2022; Lee et al., 2019) all make unrealistic assumptions. Specifically, the NTK analysis[1,5,6,7,8] is done under rather unrealistic assumptions such as extremely large width and large Gaussian initialization. Further, (Chizat et al., 2019) show that the performance of neural networks degrades when trained in the NTK regime, thus the NTK analysis does not capture the behavior of neural networks used in practice. On the other hand, (Arora et al., 2018; Du et al., 2018a) focus on deep linear networks under the assumption that there is sufficient margin at initialization. This requires the initialization to be close to optimal, which is rarely seen in practice. Therefore, both lines of work make some unrealistic assumptions. The goal of our work is precisely to relax these unrealistic assumptions by considering general initializations with either sufficient margin or sufficient imbalance, which subsumes NTK initialization, mean-field initialization and spectral initialization. Moreover, our work assumes the width is larger than or equal to the input and output dimensions, which can be considered as a "mild-overparameterization" compared with the NTK analysis. Therefore, our work is an important stepping stone to analyzing deep networks of finite width under general initialization, which is not covered by existing results.
>
> [1]Nguyen, Quynh N., and Marco Mondelli. "Global convergence of deep networks with one wide layer followed by pyramidal topology." Advances in Neural Information Processing Systems 33 (2020): 11961-11972.
>
> [2]Radhakrishnan, Adityanarayanan, Mikhail Belkin, and Caroline Uhler. "Overparameterized neural networks implement associative memory." Proceedings of the National Academy of Sciences 117.44 (2020): 27162-27170.
>
> [3]Arora, Sanjeev, et al. "A convergence analysis of gradient descent for deep linear neural networks." arXiv preprint arXiv:1810.02281 (2018).
>
> [4]Du, Simon S., Wei Hu, and Jason D. Lee. "Algorithmic regularization in learning deep homogeneous models: Layers are automatically balanced." Advances in neural information processing systems 31 (2018).
>
> [5]Du, Simon S., et al. "Gradient descent provably optimizes over-parameterized neural networks." arXiv preprint arXiv:1810.02054 (2018).
>
> [6]Jacot, Arthur, Franck Gabriel, and Clément Hongler. "Neural tangent kernel: Convergence and generalization in neural networks." Advances in neural information processing systems 31 (2018).
>
> [7]Liu, Chaoyue, Libin Zhu, and Mikhail Belkin. "Loss landscapes and optimization in over-parameterized non-linear systems and neural networks." Applied and Computational Harmonic Analysis 59 (2022): 85-116.
>
> [8]Lee, Jaehoon, et al. "Wide neural networks of any depth evolve as linear models under gradient descent." Advances in neural information processing systems 32 (2019).
>
> [9]Chizat, Lenaic, Edouard Oyallon, and Francis Bach. "On lazy training in differentiable programming." Advances in neural information processing systems 32 (2019).

---

### Official Review · Reviewer_m2Ce · 2023-10-31

**Soundness:** 3 good
**Presentation:** 3 good
**Contribution:** 1 poor
**Rating:** 6
**Confidence:** 4

**Summary:**

The paper studies the convergence rate of gradient descent for overparametrized two layer linear neural networks with generic loss.
It does so without assumptions previously used in the literature, be it on infinitesimal stepsize, infinite width, etc.
Instead, the analysis is based on local versions of the Polyak-Lojasiewicz inequality and of the descent lemma, where the global constants in both inequalities are replaced by iterate dependent versions (eq 10).
The analysis up to Eq 14 is straightforward, and most of the work consists in showing that there exists a choice of stepsize $\eta_t$ that can ensure $0 < (1 - 2 \mu_t \eta_t + \mu_t K_t \eta_t^2) \leq \rho < 1$ and $\eta_t K_t < 2$ simultaneously.

**Strengths:**

Apart from the work of the previous of Xu et al (2023), the paper is the first to study the setting of finite stepsize, finite width and "general" init (still requiring imbalance)

**Weaknesses:**

- There is **very limited novelty** with respect to Xu et al 2023, "Linear Convergence of Gradient Descent For Finite Width Over-parametrized Linear Networks With General Initialization". If the authors could point at the novelty in the proofs, it'd be more convincing, because they seemed extremely similar and this felt thin-sliced.
- There is still a dependency on initialization through the assumption on $\alpha_1$, which excludes some initializations.

**Questions:**

Can the authors detail the novelty in the proof compared to previous Xu work?




Minor comments:
## References
A work which "revived" the interest in PL form the Optimization community is "Linear Convergence of Gradient and Proximal-Gradient Methods Under the Polyak-Łojasiewicz Condition", Karimi 2016, which the authors could cite.


## Cosmetics:
- the way the authors cite the Descent Lemma is broken: " where Descent lemma is" should be "where the Descent Lemma is", same for "to derive Descent lemma," etc
- "for arbitrary non-convex functions that is": for *an* arbitrary non-convex *function* that is (singular)
- "satisfies μ-PL condition.": missing "the"
- "via chain rule:": missing "the"
- P6 "In §2.1, we show that as": we showed
- "if the $\lim_{t \to \infty}": extra "the", this time.
- "too larger" is incorrect; this whole paragraph has other typos ("but not too much $\eta_t \leq 1/K_t$)

---

> ### Author Response · Authors · 2023-11-23
>
> To [m2Ce], we thank the reviewer for the thoughtful comments. We address the questions below.
> 1. Incremental w.r.t. (Xu et al., 2023).
> Please see our response to all reviewers.
> 2. Verification of $\alpha_1>0$.
> Please see our response to all reviewers.
> 3. Missing reference: we thank the reviewer for pointing us to this work. We have added the reference to Karimi et. al. 2016 in the "Related Work" section.
> 4. Cosmetics: we thank the reviewer for the suggestions. We have modified the paper accordingly. For Descent lemma, we treated it as a noun and that's why we cite it as "Descent lemma" instead of "the Descent lemma".
>
> [1]Xu, Ziqing, et al. "Linear Convergence of Gradient Descent for Finite Width Over-parametrized Linear Networks with General Initialization." International Conference on Artificial Intelligence and Statistics. PMLR, 2023.

---

> > ### Comment · Reviewer_m2Ce · 2023-12-04
> >
> > I thank the authors for their response. It seems that all the other reviewers share the concern about limited novelty of the paper, which was somehow hidden. The authors clarification about the better constants in rates is legitimate. All things considered, I will change my score to 6.

---

### Official Review · Reviewer_KJ6o · 2023-11-04

**Soundness:** 2 fair
**Presentation:** 3 good
**Contribution:** 2 fair
**Rating:** 3
**Confidence:** 2

**Summary:**

In this paper, the authors tackle the challenge of analyzing the convergence of gradient descent (GD) for two-layer linear neural networks with general loss functions, relaxing previous assumptions about step size, width, and initialization. They introduce a new approach based on the Polyak-Łojasiewicz (PL) condition and Descent Lemma, demonstrating that these conditions hold locally with constants depending on the network's weights. By bounding these local constants related to initialization, current loss, and non-overparameterized model properties, the paper establishes a linear convergence rate for GD. Importantly, the study not only enhances previous results but also suggests an optimized step size choice, validated through numerical experiments. The authors further prove that local PL and smoothness constants can be uniformly bounded by specific properties of the non-overparameterized models. The paper concludes by proposing an adaptive step size scheme, accelerating convergence compared to a constant step size, and empirically validating the derived convergence rate's accuracy.

**Strengths:**

$\textbf{(1) Rigorous analysis of convergence conditions}$: A key strength of this paper is its rigorous analysis of convergence conditions for two-layer linear neural networks. The authors thoroughly explore the convergence behavior of gradient descent under various circumstances, relaxing previous assumptions and providing a detailed understanding of the impact of factors such as step size, width, and initialization. By establishing convergence conditions and deriving a linear convergence rate, the paper significantly advances the theoretical understanding of optimization processes in neural networks.

$\textbf{(2) Adaptive step size scheme}$: The paper proposes an adaptive step size scheme based on the derived convergence analysis. Introducing this adaptive approach showcases the practical implications of the research findings. By suggesting a dynamic step size strategy that accelerates convergence compared to a constant step size, the paper offers a concrete and actionable method for improving optimization efficiency in neural networks. This innovation enhances the applicability of the research, providing a valuable contribution to the field of optimization techniques for machine learning models.

**Weaknesses:**

$\textbf{(1) Incremental contribution}$:  Arora et al. in Ref [1] studied linear convergence of gradient descent for multi-layer neural networks. While Arora et al. assumed balanced weights and a deficiency margin, these conditions were proven by them in the context of gradient descent. In this work, although the authors only focus on general loss, they just study two-layer linear networks. Moreover, their convergence rate also depends on margin and imbalance. The contribution of this work is very incremental in terms of Ref [1].

[1] Arora et al., A convergence analysis of gradient descent for deep linear neural networks.

$\textbf{(2) Limited generalizability to deep linear networks}$: It seems that the authors don't mention how to generalize their results to deep linear networks. It is believed that deep networks are more commonly used in applications. The paper leaves a significant gap in its discussion by omitting details on the generalization of their findings to deep linear networks.

**Questions:**

$\textbf{Q1.}$ Is it possible to generalize the current analysis to deep linear networks or deep nonlinear networks? T

$\textbf{Q2.}$  In Theorem 3.2, it is assumed that $\alpha_1 > 0$. Can the authors verify this condition?

---

> ### Author Response · Authors · 2023-11-23
>
> To [KJ6o]: we thank the reviewer for the thoughtful comments. We address the questions below.
> 1. Incremental contribution compared with (Arora et al., 2018).
> We respectfully disagree that our work is an incremental contribution compared with (Arora et al., 2018). In the work of (Arora et al., 2018), the authors provide the convergence rate of deep linear networks trained with gradient descent. While deep linear networks are indeed more general, (Arora et al., 2018) focus on deep linear networks under the assumption that there is sufficient margin and small imbalance at initialization. This requires the initialization to be close to optimal, which is rarely seen in practice. Moreover, standard Gaussian initialization, Xavier initialization, and He initialization all have large imbalance at initialization. Therefore, the convergence in (Arora et al., 2018) is proved under some unrealistic assumptions that cannot be satisfied by practical initialization schemes. The goal of our work is precisely to relax these unrealistic assumptions by considering general initialization with either sufficient margin or sufficient imbalance. Therefore, our work is an important stepping stone to analyzing deep networks of finite width under general initialization.
> 2. Generalizability to deep linear networks.
> Extending the results of this work to deep linear networks trained via gradient descent is the next step, and the analysis can be done in the same spirit as it was done in this work. In the case of two-layer linear networks, we present a framework showing that one can use the imbalance and margin to bound the singular values of \mathcal{T} during the training. In the case of deep linear networks, $\mathcal{T}$ takes a different form depending on all layer weights which is more complicated than the two-layer case. On the other hand, we have $L-1$ imbalances for L-layer deep linear networks. As long as one can bound the $\mathcal{T}$ for deep linear networks using $L-1$ imbalances and the margin, the convergence rate of deep linear networks is derived. We refer the reviewer to (Min et al., 2023) which proves the convergence of deep linear networks trained via gradient flow in a similar spirit as described above.
> 3. Verification of $\alpha_1>0$.
> Please see our response to all reviewers.
>
> [1]Arora, Sanjeev, et al. "A convergence analysis of gradient descent for deep linear neural networks." arXiv preprint arXiv:1810.02281 (2018).
>
> [2] Min, H., Vidal, R. and Mallada, E., 2023. On the Convergence of Gradient Flow on Multi-layer Linear Models.

---

### Official Review · Reviewer_PUum · 2023-11-06

**Soundness:** 3 good
**Presentation:** 3 good
**Contribution:** 2 fair
**Rating:** 5
**Confidence:** 3

**Summary:**

The paper studies convergence of gradient descent for matrix factorization (also called over-parametrized linear models). Prior work[1] established linear convergence for the quadratic loss by introducing two constants ($c_1, c_2$) to bound changes in singular values along trajectory of GD. This work writes their result for stronlgy convex and smooth losses and tracks those changes better which allows for an easier computation of adaptive stepsizes.

---
[1]Xu, Ziqing, et al. "Linear Convergence of Gradient Descent for Finite Width Over-parametrized Linear Networks with General Initialization." International Conference on Artificial Intelligence and Statistics. PMLR, 2023.

**Strengths:**

- This work cleans up and offers an improved analysis of a prior result.
- It is clearly written.

**Weaknesses:**

- The prior work this works improves on already has entire sections on adaptive step sizes: It is in a small sentence on page 8 we discover that [1] already proposes adaptive stepsizes when throughout it is presented as only having fixed stepsize schemes. The presentation of the prior work needs to include this fact.
- Significance: This result is carefully analyzes matrix factorization, after papers before have proved linear convergence of GD. Once the linear convergence question has been answered, can the authors justify why it is still significant to study matrix factorization ? The original reason for studying this simplified setting was to prove that non-convexity can be benign. This question was already answered. So the authors should provide more arguments as to why it would still be interesting to derive adaptive stepsizes to improve an already linear rate.

**Questions:**

- Would the authors agree to say that the central contribution of this work that differentiates it from [1] is lemma 3.1 ?

---

> ### Author Response · Authors · 2023-11-23
>
> To [PUum]: we thank the reviewer for the thoughtful comments. We address the question below.
>
> 1. Is Lemma 3.1 the main contribution of this work?
> We respectfully disagree that Lemma 3.1 is the main (technical) contribution of this work compared with (Xu et al., 2023). The main technical contributions of this work consist of (a) a novel local smoothness inequality and Descent lemma (See Theorem 3.1), (b) a tighter convergence rate for overparametrized models with a general loss (See Theorem 3.2), and (b) novel spectral bounds for $\mathcal{T}$ (See Lemma 3.1). Specifically, Theorem 3.1 presents a novel local Descent lemma and PL inequality of the overparametrized model to characterize the local rate of decrease, while prior results (Arora et al., 2018; Du et al., 2018a; Xu et al., 2023) are based on Descent lemma and PL inequality of the non-overparametrized model. In Appendix C, we show that Theorem 3.1 leads to a faster rate of decrease compared with the results in (Arora et al., 2018; Du et al., 2018a; Xu et al., 2023). Based on Theorem 3.1, it suffices to bound the singular values of \mathcal{T} and W to derive the linear convergence of the overparametrized model. Lemma 3.1, which is the second main contribution of this work, shows when the step size satisfies certain constraints that allow the step sizes to grow during the training, one can present uniform spectral bounds for $\mathcal{T}$ and W using the initial weights and initial step size. In (Xu et al., 2023), the authors show that the uniform spectral bounds not only depend on initial weights but also on some auxiliary constants $c_1, c_2$ under constant upper bound on the step sizes. Thus, the requirement in Lemma 3.1 admits a wider choice of the step sizes. Moreover, (Xu et al., 2023) show linear convergence under restrictive assumptions on $c_1$ and $c_2$, which leads to looser uniform spectral bounds on $\mathcal{T}$ compared with this work. Finally, Theorem 3.2, which is based on Theorem 3.1 and Lemma 3.1, shows linear convergence of the adaptive step size scheme. In Appendix G, we numerically verify that the adaptive step size scheme proposed in this work is almost ten times faster than the one proposed in (Xu et al., 2023) and the backtracking line search.
>
> 2. Significance of studying matrix factorization.
> Please see our response to all reviewers.
>
> 3. Improvement of the presentation of the prior work.
> We thank the reviewer for the suggestion. We modified the "Related Work" section to highlight that (Xu et al., 2023) also designed an adaptive step size scheme to accelerate the convergence, and the convergence rate of the adaptive step size scheme heavily relies on two auxiliary constants which is a slower rate compared with this work.
>
> [1]Xu, Ziqing, et al. "Linear Convergence of Gradient Descent for Finite Width Over-parametrized Linear Networks with General Initialization." International Conference on Artificial Intelligence and Statistics. PMLR, 2023.

---

### Author Response · Authors · 2023-11-23

We thank all the reviewers for their thoughtful comments and address their common questions below. We also uploaded a new version of the paper, where we highlighted the changes made in blue.

(PUum, KJ6o)Significance of this work: We would like to highlight the significance of our work compared with existing literature.
Our work not only proves the linear convergence of GD for two-layer linear networks for general loss functions but also provides (a) a new conceptual framework that allows us to understand the effect of overparametrization (expressed via the operator $\mathcal{T}$) in convergence, and (b) a learning rate schedule that leads to almost optimal convergence rate.
While prior work (Arora et al., 2018; Xu et al., 2023) provides linear rates for GD, this does not mean that overparametrization has a benign effect, given the fact that previously proposed rates can be quite far from the optimal non-overparametrized rate of $1-\frac{\mu}{K}$. The state-of-the-art (asymptotic) rate, before our work, was provided in (Xu et al., 2023), and in our notation, amounts to $\left(1-\left(\frac{c_1\alpha_1}{c_2\alpha_2}\right)^2\frac{\mu}{K}\right)$. Our work, in comparison, archives a rate of $1-\frac{\alpha_1}{\alpha_2}\frac{\mu}{K}$.
Moreover, in Appendix E of Xu et al., 2023, the authors show their characterization of the local convergence rate holds under stringent assumptions on $c_1, c_2$, i.e., $\frac{c_1}{c_2} << 1$.
Even in a mild case which is the same setting as the simulation done in this work with $h=1000, \sigma^2=0.1$. The theory in this work and (Xu et al., 2023) (See Claim E.1) yields $\frac{c_1}{c_2}=1/3, \frac{\alpha_1}{\alpha_2}=0.6746, \frac{\mu}{K}=1.0$, our analysis gives a significantly better rate (1-0.6746) when compared with Xu et al. 2023 (1-0.0506).
Notably, $\frac{\alpha_1}{\alpha_2}$, which in our work is associated with the conditioning of the operator $\mathcal{T}$, can be controlled with proper initialization, thus leading to an almost optimal rate! To our knowledge, this is the first work to achieve nearly optimal rates.
Our improvement is also evident from the experiments where we show that the adaptive step size scheme proposed in this work improves the convergence rate and is numerically almost ten times faster than the one presented in (Xu et al., 2023) and backtracking line search (See Appendix G). We further note that the values of $c_1$ and $c_2$ chosen in this comparison are smaller than the ones prescribed by the theory of (Xu et al., 2023).

(PUum, KJ6o, m2Ce)Incremental w.r.t. (Xu et al., 2023): We respectfully disagree that our work seems incremental and less surprising compared to [1]. We added, in the revision, a discussion about the difference in analysis techniques between this work and (Xu et al., 2023) in Appendix C in detail. Specifically, we consider general strongly convex and smooth losses in this work while (Xu et al., 2023) only consider the squared loss. Moreover, we show that the convergence rate in this work is one condition number of $\mathcal{T}$ better than the one derived in prior work (Arora et al., 2018; Du et al., 2018a; Xu et al., 2023) (See Appendix C for detail). Finally, we propose an adaptive step size scheme that is more computationally efficient compared with (Xu et al., 2023). In Appendix G, we numerically verify that the adaptive step size scheme proposed in this work is almost ten times faster than the one proposed in (Xu et al., 2023) and backtracking line search.

(KJ6o, m2Ce, 7VmP)Restrictive Assumption $\alpha_1>0$: We presented two conditions that ensure $\alpha_1>0$ in Appendix F. Specifically, the first condition states that for Gaussian initializations with proper choices of the variance, when the model is "sufficiently" overparametrized $\alpha_1$ has a positive lower bound with high probability over the random initialization. Moreover, the lower bound increases as the width, i.e., h, increases. The other condition states for any initialization scheme that sample weights from a continuous distribution, as long as $h \geq m+n$, we almost surely, i.e. with probability 1, have $\alpha_1>0$. We refer the reviewers to Appendix F for details. Thus, all commonly used initialization schemes, such as Gaussian initialization, He initialization, and Xavier initialization, have $\alpha_1>0$ when h $\geq m+n$.

[1]Xu, Ziqing, et al. "Linear Convergence of Gradient Descent for Finite Width Over-parametrized Linear Networks with General Initialization." International Conference on Artificial Intelligence and Statistics. PMLR, 2023.

[2]Arora, Sanjeev, et al. "A convergence analysis of gradient descent for deep linear neural networks." arXiv preprint arXiv:1810.02281 (2018).

[3]Du, Simon S., Wei Hu, and Jason D. Lee. "Algorithmic regularization in learning deep homogeneous models: Layers are automatically balanced." Advances in neural information processing systems 31 (2018).

---

### Meta-Review · Area_Chair_opDx · 2023-12-10

**Metareview:**

Existing results on the convergence of gradient descent (GD) in overparameterized neural networks have often depended on stringent assumptions about the step size (being infinitesimal), the width of hidden layers (being infinite), or the initialization (being large, spectrally balanced). In this paper building upon prior work, the authors claim to establish a linear convergence rate for training two-layer linear neural networks using gradient descent that is applicable to general losses and under more relaxed criteria concerning step size, network width, and initialization compared to prior work.

The reviewers thought the paper provides a rigorous analysis of convergence conditions and like the adaptive step size scheme but thought that the contributions to be somewhat incremental and unlikely to extend to deep linear networks. The authors' response did not alleviate these concerns. I concur with most of the reviewers while there are some interesting analysis here, overall the contributions do not rise to the level that is suitable for publication at ICLR.

**Justification For Why Not Higher Score:**

the paper IMO is incremental and the reviewers seem to concur

**Justification For Why Not Lower Score:**

N/A

---

### Decision · Program_Chairs · 2024-01-16

Reject